# A simple mechanism for integration of quorum sensing and cAMP signalling in *Vibrio cholerae*

Lucas M Walker[1], James RJ Haycocks[1], Julia C Van Kessel[2], Triana N Dalia[2], Ankur B Dalia[2], David C Grainger[1]*

[1]School of Biosciences, University of Birmingham, Edgbaston, United Kingdom; [2]Department of Biology, Indiana University, Bloomington, United States

*For correspondence:
d.grainger@bham.ac.uk

Competing interest: The authors declare that no competing interests exist.

**Abstract** Many bacteria use quorum sensing to control changes in lifestyle. The process is regulated by microbially derived 'autoinducer' signalling molecules, that accumulate in the local environment. Individual cells sense autoinducer abundance, to infer population density, and alter their behaviour accordingly. In *Vibrio cholerae*, quorum-sensing signals are transduced by phosphorelay to the transcription factor LuxO. Unphosphorylated LuxO permits expression of HapR, which alters global gene expression patterns. In this work, we have mapped the genome-wide distribution of LuxO and HapR in *V. cholerae*. Whilst LuxO has a small regulon, HapR targets 32 loci. Many HapR targets coincide with sites for the cAMP receptor protein (CRP) that regulates the transcriptional response to carbon starvation. This overlap, also evident in other *Vibrio* species, results from similarities in the DNA sequence bound by each factor. At shared sites, HapR and CRP simultaneously contact the double helix and binding is stabilised by direct interaction of the two factors. Importantly, this involves a CRP surface that usually contacts RNA polymerase to stimulate transcription. As a result, HapR can block transcription activation by CRP. Thus, by interacting at shared sites, HapR and CRP integrate information from quorum sensing and cAMP signalling to control gene expression. This likely allows *V. cholerae* to regulate subsets of genes during the transition between aquatic environments and the human host.

## eLife assessment

This paper provides **valuable** new information on the mechanisms by which *Vibrio cholerae* integrates and responds to environmental signals. The strength of the evidence provided in support of the conclusions made and the model proposed is **solid**. The revision resolved many of the issues raised by the reviewers and improved the manuscript. The work is relevant for a broad audience of microbiologists interested in the mechanisms by which bacteria sense their environment.

## Introduction

*Vibrio cholerae* is a Gram-negative bacterium responsible for the human disease cholera (***Nelson et al., 2009***). Estimates suggest 3 million annual infections, of which 100 thousand are fatal (***Ali et al., 2015***). Most disease instances are attributed to the El Tor *V. cholerae* biotype, which is responsible for the ongoing 7th cholera pandemic (***Domman et al., 2017***). Globally, over 1 billion people inhabit areas of endemicity and future climatic change is likely to exacerbate the risk of illness (***Ali et al., 2015***; ***Asadgol et al., 2019***). The success of *V. cholerae* as a pathogen is underpinned by an ability to colonise both aquatic ecosystems and the human intestinal tract (***Nelson et al., 2009***). In waterways, *V. cholerae* prospers by forming biofilms on arthropod exoskeletons. Degradation of these

chitinous surfaces ultimately liberates *N*-acetylglucosamine (GlcNAc) for metabolism by the microbe (*Meibom et al., 2004*). Upon ingestion by a human host, *V. cholerae* express genetic determinants for acid tolerance, intestinal colonisation, and virulence. Diverse transcription factors regulate the transition and respond to signals including bile (*Hung and Mekalanos, 2005*), temperature (*Weber et al., 2014*), nucleotide second messengers (*Krasteva et al., 2010*; *Manneh-Roussel et al., 2018*), and chitin availability (*Meibom et al., 2004*). Understanding these regulatory networks is important to determine how *V. cholerae* can switch between environments to cause disease outbreaks (*Domman et al., 2017*; *Kamareddine et al., 2018*; *Weill et al., 2017*).

Quorum sensing is key for the transition of *V. cholerae* between ecological niches (*Eickhoff and Bassler, 2018*). Briefly, *V. cholerae* produce at least 3 autoinducer (AI) signalling molecules: cholera AI-1 (CAI-1), AI-2, and 3,5-dimethylpyrazin-2-ol (DPO) (*Mukherjee and Bassler, 2019*). In the environment, these compounds are detected by receptors in neighbouring cells and indicate population density. Importantly, whilst AI-2 and DPO are produced by multiple bacterial species, CAI-1 is only made by other members of the *Vibrio* genus (*Henke and Bassler, 2004*). Thus, *V. cholerae* can determine the crude composition of bacterial populations. In the absence of their cognate AIs, when population density is low, the receptors for CAI-I and AI-2 target the transcription factor LuxO for phosphorylation via a phosphorelay system (*Mukherjee and Bassler, 2019*; *Freeman and Bassler, 1999a*; *Freeman and Bassler, 1999b*). When phosphorylated, LuxO upregulates the production of four small quorum regulatory RNAs (Qrrs) (*Lenz et al., 2004*). In turn, the Qrrs control expression of two global transcription factors: AphA and HapR (*Lenz et al., 2004*; *Shao and Bassler, 2012*; *Rutherford et al., 2011*). Importantly, whilst AphA production is activated by Qrrs, synthesis of HapR is repressed. Hence, AphA and HapR control gene expression at low and high cell density respectively (*Mukherjee and Bassler, 2019*; *Rutherford et al., 2011*). A simplified outline of the LuxO dependent regulatory pathway for HapR is illustrated in *Figure 1a*.

Identified as a regulator of *hapA*, required for *V. cholerae* migration through intestinal mucosa, HapR is a TetR-family member that binds DNA as a homodimer via a N-terminal helix-turn-helix motif (*De Silva et al., 2007*; *Jobling and Holmes, 1997*). Many clinical isolates of pandemic *V. cholerae* have lost the ability to properly express HapR and this may indicate adaptation to a more pathogenic lifestyle (*Domman et al., 2017*; *Kamareddine et al., 2018*; *Heidelberg et al., 2000*). In *V. cholerae*, HapR regulates the expression of ~100 genes to promote 'group behaviours' including natural competence, repression of virulence genes, and escape from the host intestinal mucosa (*Nielsen et al., 2006*). In other *Vibrio* spp., equivalent regulons are larger. For example, LuxR in *Vibrio harveyi* regulates over 600 genes (*van Kessel et al., 2013a*). Expression of HapR can be influenced by other factors. In particular, cAMP receptor protein (CRP), a regulator that controls metabolism of alternative carbon sources, including chitin, upregulates HapR (*Silva and Benitez, 2004*). In this study, we used chromatin immunoprecipitation and DNA sequencing (ChIP-seq) to identify direct DNA binding targets of HapR and its upstream regulator, LuxO. We show that the degenerate DNA consensus bound by HapR frequently overlaps targets for CRP. At such sites, HapR and CRP co-operatively bind offset faces of the double helix. Strikingly, this occludes a key CRP surface required to activate transcription. This simple mechanism allows *V. cholerae* species to integrate quorum sensing, and cAMP signalling, in the control of gene expression.

## Results

### Genome-wide DNA binding by HapR and LuxO in *Vibrio cholerae*

Whilst the impact of HapR on global gene expression in *V. cholerae* has been investigated, it is not known which HapR responsive genes are directly controlled by the protein (*Nielsen et al., 2006*). Similarly, the extent of the direct LuxO regulon is unknown. Hence, we sought to map the binding of LuxO and HapR across the *V. cholerae* genome. To facilitate this, *luxO* and *hapR* were cloned in plasmids pAMCF an pAMNF respectively. The resulting constructs, encoding LuxO-3xFLAG or 3xFLAG-HapR, were used to transform *V. cholerae* strain E7946. In subsequent ChIP-seq experiments, anti-FLAG antibodies were used to select fragments of the *V. cholerae* genome bound with either LuxO or HapR. The derived binding profiles are shown in *Figure 1b*. In each plot, genes are shown as blue lines (outer two tracks) whilst the LuxO and HapR binding signals are red and green respectively (inner two tracks). Examples of individual binding peaks for each factor are shown in *Figure 1c*. In total, we identified 5

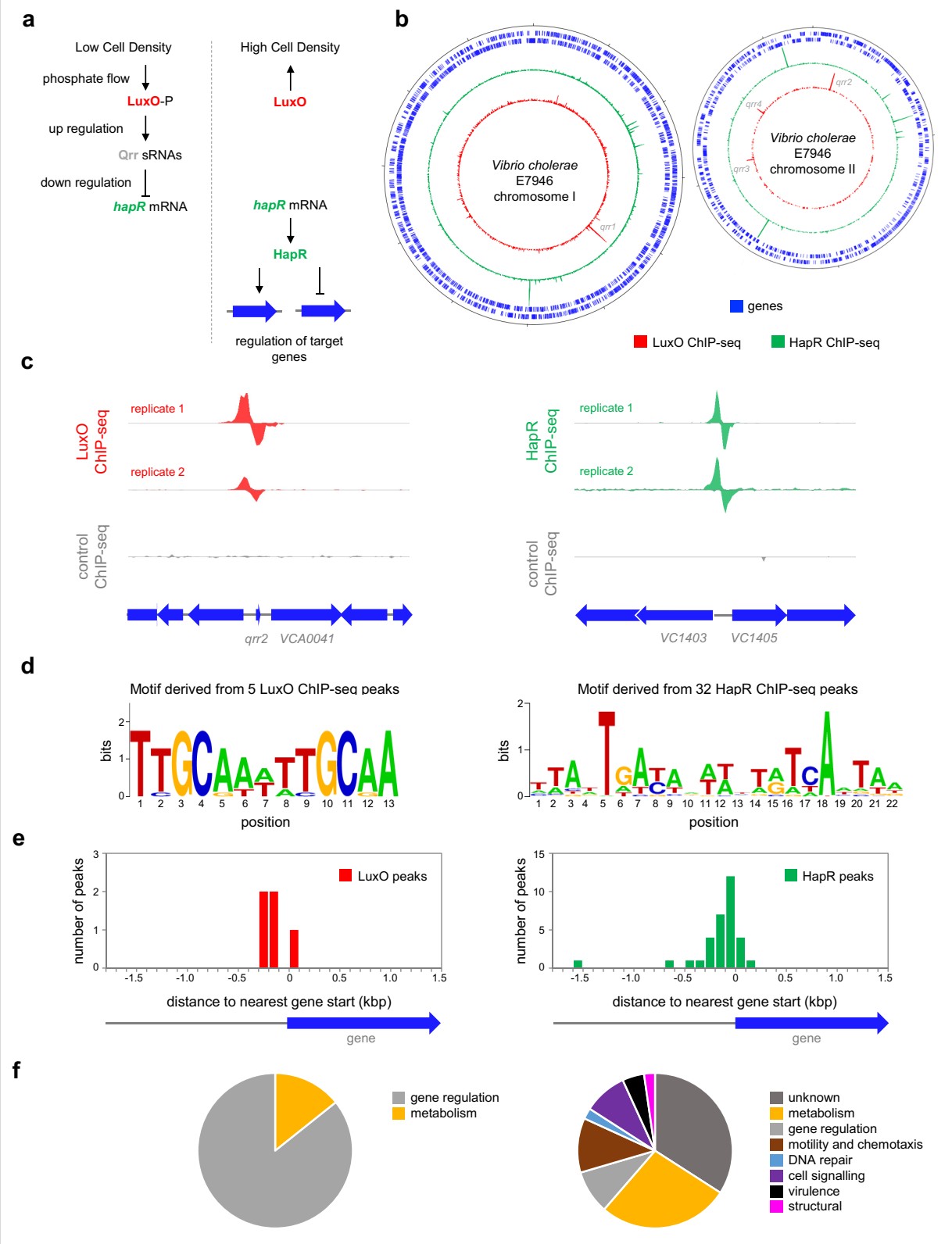

**Figure 1.** Genome-wide distribution of HapR and LuxO in *Vibrio cholerae*. (**a**) Simplified schematic overview of quorum sensing in *Vibrio cholerae*. At low cell density, expression of HapR is repressed by the Qrr sRNAs that depend on phosphorylated LuxO for activation of their transcription. Arrows indicate activation and bar ended lines indicate repression. For clarity, not all protein factors involved in the cascade have been included. (**b**) Binding of LuxO and HapR across both *Vibrio cholerae* **chromosomes**. In each plot the outer two tracks (blue) are genes orientated in the forward or reverse

*Figure 1 continued on next page*

*Figure 1 continued*

direction. The LuxO and HapR ChIP-seq binding signals are shown in red and green. LuxO binding peaks corresponding to the *qrr1-4* loci are indicated. Tick marks are 0.25 Mbp apart. (**c**) Example LuxO and HapR ChIP-seq binding peaks. ChIP-seq coverage plots are shown for individual experimental replicates. Data for LuxO and HapR are in green and red respectively. Signals above or below the horizontal line correspond to reads mapping to the top or bottom strand respectively. Gene are show as block arrows. (**d**) Sequence motifs derived from LuxO and HapR binding peaks using MEME. (**e**) Positions of LuxO and HapR binding peaks with respect to genes. The histograms show the distribution of binding peak centres with respect to the start codon of the nearest gene. (**f**) Pie charts showing gene classes targeted by LuxO and HapR.

The online version of this article includes the following figure supplement(s) for figure 1:

**Figure supplement 1.** Binding of LuxO and the *qrr1* and *VC1142* loci.

**Figure supplement 2.** Example HapR binding signals.

and 32 peaks for LuxO and HapR binding respectively (*Table 1*). Previous work identified targets for LuxO adjacent to genes encoding the 4 Qrr sRNAs. We recovered all of these known LuxO targets, and an additional binding site was identified between *VC1142* and *VC1143*. These divergent genes encode cold shock-like protein CspD, and the Clp protease adaptor protein, ClpS, respectively. Note that the LuxO binding signal at this locus is small, compared to the *qrr*1-4 targets, but may still be involved in transcription regulation (*Figure 1—figure supplement 1*). To identify the sequence bound by LuxO, DNA regions overlapping LuxO binding peaks were inspected using MEME. The motif identified matches the known consensus for LuxO binding and was found at all LuxO targets (*Table 1* and *Figure 1d*; *Tu and Bassler, 2007*). The positions of LuxO binding sites with respect to genes, and the functions encoded by these genes, are summarised in *Figure 1e and f* respectively.

Of the 32 peaks for HapR binding, 4 correspond to previously identified direct targets [*hapR* (*Lin et al., 2005*), *VC0241* (*Tsou et al., 2009*), *VC1851* (*Waters et al., 2008*) and *VCA0148* (*Tsou et al., 2009*)]. However, some known targets had weak or poorly reproducible binding signals (*Figure 1—figure supplement 2*). A DNA motif common to all 32 HapR ChIP-seq peaks matched prior descriptions of the DNA target for HapR or closely related proteins (*Lin et al., 2005*; *van Kessel et al., 2013b*; *Zhang et al., 2021*; *Figure 1d*). Occurrences of this HapR binging motif were most frequent in the 200 bp preceding a gene start codon (*Figure 1e*). Most often, the genes adjacent to HapR binding peaks encode protein functions related to metabolism, motility, and chemotaxis (*Figure 1f*). Overall, our data suggest that LuxO primarily regulates gene expression via the 4 Qrr sRNA molecules. Conversely, the genome-wide distribution of HapR is consistent with that of a global gene regulator with many undefined regulatory roles.

## HapR is a direct regulator of transcription at many target sites

We focused our attention on new HapR target promoters where adjacent coding sequence could be used to predict encoded protein function. For these 24 targets, regulatory DNA was cloned upstream of *lacZ* in plasmid pRW50T. Recombinants were then transferred to *V. cholerae* E7946, or the ΔhapR derivative, by conjugation. Strains generated were cultured overnight before β-galactosidase activities were determined. The results are shown in *Figure 2a*. Promoters were categorised as inactive, unresponsive, repressed or activated by HapR. We identified 2 and 7 promoters subject to activation and repression by HapR, respectively. Of the remaining promoters, 6 were inactive and 9 unresponsive to HapR in our conditions. Next, the 9 promoter DNA fragments responsive to HapR in vivo were cloned upstream of the $\lambda$ *oop* terminator in plasmid pSR. The resulting constructs were then provided to housekeeping *V. cholerae* RNA polymerase, as templates for in vitro transcription, in the presence and absence of HapR. The results are shown in *Figure 2b* where the expected size of transcripts terminated by $\lambda$ *oop* are marked with blue triangles (*Papenfort et al., 2015*). Recall that the *VC1375* and *VC1403* promoters were activated by HapR in vivo (*Figure 2a*). Consistent with this, HapR also activated the *VC1375* promoter in vitro (*Figure 3b*, lanes 43–47). However, HapR did not activate in vitro transcription from the *VC1403* promoter (*Figure 3b*, lanes 48–53). Indeed, interpretation of these data were hampered because the location of the *VC1403* transcription start site (TSS) is not known (*Papenfort et al., 2015*). Of the 7 promoters repressed by HapR in vivo, we observed repression in six cases in vitro (*hapR, VC0585, VC2352, VCA0219, VCA0663,* and *VCA0960*) (lanes 7–42). Conversely, the *murQP* promoter (P*murQP*) subject to repression by HapR in vivo, generated no transcript in vitro (lanes 1–6). Full gel images are shown in *Figure 2—source data 7*.

**Table 1.** Locations of binding peaks from ChIP-seq experiments.

| peak centre | gene(s)* | site location | site sequence | TSS[†] |
|---|---|---|---|---|
| | | HapR ChIP-seq peaks | | |
| | | *chromosome I* | | |
| 99874 | VC0102<(VC0103) | 99863.5 | aaattaataaaactgtcattta | 99906 (+) |
| 213457 | (VC0205)>VC0206 | 213452.5 | taattgtgattcttatcaccaa | 213494 (+) |
| 246366 [‡] | VC0240<>VC0241 | 246349.5 | taattaagatggctataaacta | 246430 (-) |
| 463584 | VC0433 | | | |
| 514422 | VC0484 | 514430.5 | ctactgaccttttcatcaataa | 514427 (+) |
| 516570 | (VC0486) | 516601.5 | caactgagaaggcacacaatag | 516545 (+) |
| 534714 | (VC0502) | 534691.5 | ctattataagctctatcagtgt | 534805 (-) |
| 547108 | VC0515 | 547135.5 | atagtaatattattgttaatag | 549431 (-) |
| 613328 [§] | VC0583[A] | 613357.5 | ttattgagtgggtacataacaa | 613427 (+) |
| 716707 | VC0668 | 716625.5 | ctattgatgaggttatccacag | 716537 (-) |
| 735309 | VC0687<>VC0688 | | | |
| 882854 | (VC0822) | 882825.5 | taattatccactttatcaattg | 883072 (-) |
| 941187 | VC0880 | 941164.5 | cttttgacatttctgtcacaaa | 941152 (+) |
| 978577 | VC0916[R] | 978540.5 | taattaatatccagctcaatta | 978581 (+) |
| 1356743 | VC1280<>VC1281[A] | 1356736.5 | atattgatagaaataacaagtc | 1356896 (+) |
| 1379202 | VC1298<>VC1299 | 1379180.5 | ttcatgatagttttgtaattat | 1379189 (+) |
| 1469384 | VC1375<>VC1376 | 1469377.5 | atattgatatatcacacatctt | 1469374 (+) |
| 1496023 | VC1403[A]<>VC1405 | 1496025.5 | tagttgatattttataattgt | 1495942 (+) |
| 1533842 | (VC1437) | 1533854.5 | tttgtgagtctcctgtcaataa | 1533703 (-) |
| 1990133 [¶] | VC1851 | 1990076.5 | atattgagtaatcaattagtaa | 1990031 (+) |
| 2364721 | (VC2212) | 2364680.5 | ctattaacagttttatttataa | 2364774 (+) |
| 2509878 | VC2352 | 2509882.5 | ttagtgacagatgcgtcattaa | 2509790 (-) |
| 2667349 | VC2486 | 2667368.5 | taattattaatttgaacaatag | 2667206 (-) |
| | | *chromosome II* | | |
| 163808 | VCA0148 | 163810.5 | taattgattattgtgtaactat | 163852 (-) |
| 214589 | (VCA0198) | 214582.5 | taattgataactttgacagtat | 213494 (+) |
| 237008 | VCA0218<>VCA0219[R] | 237019.5 | taaataatatgaatatcagtaa | 237053 (+) |
| 247286 | VCA0224<>VCA0225 | 247241.5 | taaatgactaataagacaatta | 247165 (-) |
| 598444 | VCA0691[A] | 598403.5 | tttgtaataaatttgtcattaa | 598413 (+) |
| 630517 | VCA0691[A] | 630559.5 | ctattaacaggactgacattaa | 631303 (+) |
| 862737 | VCA0906 | | | |
| 910196 | VCA0960[R]<>VCA0961 | 910181.5 | ctgattataaatttgtaaatat | 910330 (+) |
| 1021174 | VCA1070 | 1021117.5 | ctcctatccgattggtcactat | 1021326 (+) |
| | | LuxO ChIP-seq peaks | | |
| | | *chromosome I* | | |
| 1090129 | qrr1<>VC1021 | 1090154 | ttgcaaaatgcaa | 1090182 (+) |
| 1212442 | VC1142<>VC1143 | 1212435 | ttgcaaatcgcga | 1212403 (-) |

*Table 1 continued on next page*

*Table 1 continued*

| peak centre | gene(s)* | site location | site sequence | TSS† |
|---|---|---|---|---|
| | | *chromosome II* | | |
| 48415 | qrr2 | 48347 | ttgcaatttgcaa | 48851 (-) |
| 772208 | qrr3 | 772149 | ttgcattttgcaa | 772227 (+) |
| 908445 | qrr4 | 908436 | ttgcaatttgcaa | 908475 (+) |

*Identified as activated (A) or repressed (R) by **Nielsen et al., 2006**, VC0206, VC0240, VC0241, VC0583, VC0668, VC0916, VC1021, VC1142, VC1143, VCA0219 correspond to *murQ*, *rfaD*, *rfbA*, *hapR*, *mutH*, *vpsU*, *luxO*, *cspD*, *clpS*, *hlyA* respectively.

†Nearest transcription start site (TSS) identified by **Papenfort et al., 2015** with the symbol in parenthesis indicating the direction of transcription. Note that these are not necessarily those TSSs subject to regulation by HarR or LuxO, particularly if the regulators are acting as repressors, or if the gene subject to regulation is switched off for another reason in the conditions of Papenfort et al.

‡Identified by Tsou and co-workers (**Tsou et al., 2009**).

§Identified by Lin and co-workers (**Lin et al., 2005**).

¶Identified by Waters and co-workers (**Waters et al., 2008**).

### Transcription from the murQP *promoter requires CRP* in vivo *and* in vitro

The *murQP* operon (*VC0206-VC0207*) encodes functions important for recycling of peptidoglycan (**Borisova et al., 2016**). Briefly, cell wall derived *N*-acetylmuramic acid (MurNAc) is transported across the inner membrane, and simultaneously phosphorylated, by the phosphotransferase system dependent permease MurP. Resulting MurNAc-6P is hydrolysed by MurQ to generate *N*-acetylglucosamine 6-phosphate (GlcNAc-6P). Intriguingly, GlcNAc-6P can also be derived from chitin break down and this coincides with expression of HapR. Hence, we focused on understanding the role of HapR bound upstream of *murPQ*. The HapR ChIP-seq binding signal at the *murQP* locus is shown in **Figure 3a** and the associated regulatory region is shown in **Figure 3b**. The centre of the ChIP-seq peak for HapR is marked by an asterisk and the predicted binding site is highlighted green. We reasoned that our inability to detect transcription from P*murQP* in vitro was likely because an undefined transcriptional activator is absent (**Figure 2b**). Inspection of the DNA sequence upstream of *murQP* identified a close match to the consensus binding site for CRP (5'-TGTGA-N$_6$-TCACA-3'). Furthermore, this sequence was located 41.5 bp upstream, of the *murQP* TSS (**Figure 3b**). This is a common scenario for CRP dependent transcription activation (**Savery et al., 1998**). To measure binding of CRP to the *murQP* regulatory region we used electrophoretic mobility shift assays (EMSAs). Consistent with our prediction, CRP bound to the *murQP* regulatory DNA (**Figure 3c**, lanes 1 and 2). To confirm that we had correctly identified the binding site for CRP, we made a series of P*murQP* derivatives. The Δ183 and Δ211 DNA fragments have large upstream deletions (sites of truncation are shown by inverted triangles in **Figure 3b**, which mark the 5' end of the remaining promoter DNA) but still bind CRP (**Figure 3c**, lanes 3–6). Conversely, point mutations –35 g and –49 g, within the CRP site, prevent binding (**Figure 3c**, lanes 7–8). To determine the impact of CRP on P*murQP* activity we first used in vitro transcription assays (**Figure 3d**). Addition of CRP to reactions resulted in production of an RNA from P*murQP*. We observed similar CRP dependence in vivo using β-galactosidase assays (**Figure 3e**, compare wild type promoter activity with and without CRP). Furthermore, in wild type cells, the –35 g and –49 g mutations reduced promotor activity whilst the Δ183 and Δ211 truncations did not (**Figure 3e**). We note that the Δ211 derivative is much more active than the starting promoter DNA sequence, but transcription remains totally dependent on CRP. Most likely, the truncation removes a repressive DNA element upstream of the core promoter.

### HapR and CRP bind a shared DNA site at the murQP *promoter*

At P*murQP*, the DNA site for CRP is completely embedded within the predicted HapR binding sequence (**Figure 3b**). To better understand this unusual configuration, we used DNaseI footprinting. The results are shown in **Figure 4a**. Lane 1 shows the pattern of DNaseI digestion in the absence of bound protein. In the presence of CRP (lanes 2–4) a footprint was observed between positions –29

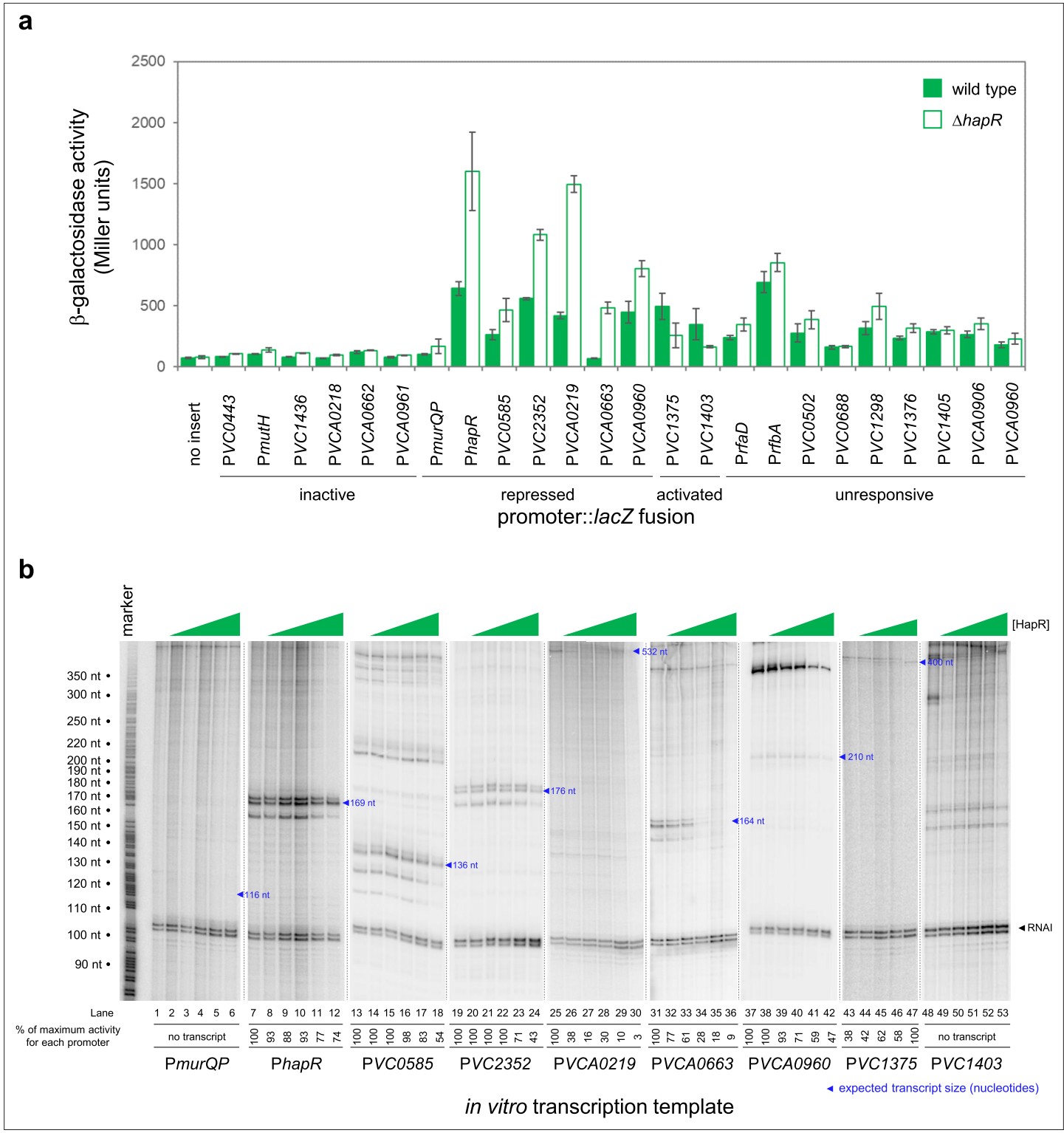

**Figure 2.** HapR is a direct repressor of transcription at many target promoters. (**a**) Activity of HapR targeted promoters in the presence and absence of HapR in vivo. The promoter regions of HapR targeted genes were fused to *lacZ* in plasmid pRW50T and constructs used to transform required bacterial strains. β-galactosidase activity was measured in cell lysates taken from *Vibrio cholerae* E7946 (bars) or the Δ*hapR* derivative (open bars). containing the *VC0857* promoter cloned upstream of *lacZ*. Standard deviation is shown for three independent biological replicates. Cells were grown in LB-Lennox medium at 37 °C to an $OD_{650}$ of ~1.1. Promoters were classified as inactive if, in both the presence and absence of HapR, β-galactosidase activity was <2 fold higher than the equivalent no insert control. We have labelled promoters with gene names or locus tags as most appropriate. Note that the *Table 1* footnote can be used to cross reference between locus tags and gene names where relevant. (**b**) Activity of HapR targeted promoters in

*Figure 2 continued on next page*

*Figure 2 continued*

the presence and absence of HapR in vitro. The gel images show results of in vitro transcription experiments. The DNA templates were plasmid pSR derivatives containing the indicated regulatory regions. Experiments were done with 0.4 μM RNA polymerase in the presence (0.25, 0.75, 1.0, 3.0, or 5.0 μM) and absence of HapR. Except for the *VC1375* promoter, where the maximum HapR concentration was 3.0 μM. The RNAI transcript is plasmid-derived and acts as an internal control. Expected transcript sizes, based on results from global transcription start site mapping experiments (*Papenfort et al., 2015*), are indicated. Note that no *VC1403* transcript was detected in this prior study (*Papenfort et al., 2015*).

The online version of this article includes the following source data for figure 2:

Source data 1. Gel image TIFF file, *Figure 2b*.
Source data 2. Gel image TIFF file, *Figure 2b*.
Source data 3. Gel image TIFF file, *Figure 2b*.
Source data 4. Gel image TIFF file, *Figure 2b*.
Source data 5. Gel image TIFF file, *Figure 2b*.
Source data 6. Gel image TIFF file, *Figure 2b*.
Source data 7. Original gel images.

and −59 bp relative to the *murQP* TSS. As is usual for CRP, and a consequence of DNA bending, the footprint comprised protection from, and hypersensitivity to, DNAse I attack. Three distinct sites of DNAseI hypersensitivity are marked by orange arrows alongside lane 4 in *Figure 4a*. The pattern of DNAse I digestion in the presence of HapR is shown in lanes 5–8. The footprint due to HapR binding exactly overlaps the region bound by CRP and results in complete protection of the DNA from digestion between positions −29 and −58 (green bar adjacent to lane 8). We also observed changes in the relative intensity of bands upstream of the HapR site between promoter positions −60 and −80. We speculate that this may result from changes in DNA conformation. Importantly, there was one further subtle difference between HapR and CRP induced banding patterns. Namely, in the presence of HapR, a band was observed at position −58 (see green triangle adjacent to lane 8). With CRP, a band was instead observed at position −59 (compare lanes 2–4 with 5–8). In a final set of assays, we examined addition of CRP and HapR in unison. We reasoned that three outcomes were possible. First, one of the two protein factors could outcompete the other. This should result in a DNAse I digestion pattern identical to either the individual CRP or HapR footprint. Second, some DNA fragments in the reaction could be bound by CRP and others by HapR. In this case, a mixed DNAse I digestion pattern, containing all features of the individual footprints due to CRP and HapR, should occur. Third, CRP and HapR could bind simultaneously. This might generate a DNAse I digestion pattern with similarities to the CRP and HapR footprints. However, accessibility of the nucleic acid to DNAse I would likely be altered in some way, with unpredictable outcomes. The result of the experiment was analysed in lanes 9–12. The binding pattern matched only some aspects of the individual footprints for CRP and HapR. Hence, we observed 2 of the 3 DNAse I hypersensitivity sites due to CRP binding. Changes in the banding pattern upstream of the binding sequence, due to HapR, were also detected. We did not observe the band at position −58 detected with HapR alone. Rather, we observed a band at position −59. An additional band at position −26 (black triangle adjacent to lane 12) was unique to these reactions. We conclude that HapR and CRP recognise the same section of the *murQP* regulatory region and may bind in unison.

### HapR and CRP bind the murQP promoter co-operatively

Fragments of the *murQP* regulatory DNA, simultaneously bound by CRP and HapR, are expected to have distinct migratory properties during electrophoresis. Thus, we compared binding of CRP and/or HapR using EMSAs. The results are shown in *Figure 4b*. As expected, addition of CRP to reactions caused a distinct shift in electrophoretic mobility (lanes 1–5). Comparatively, at the concentration used, HapR bound the DNA fragment poorly; we observed only smearing of the free DNA at the highest HapR concentration tested (lanes 6–10). The binding pattern due to HapR was dramatically different if DNA was pre-bound with CRP (lanes 11–15). In this scenario, even low concentrations of added HapR were sufficient to generate a super-shifted nucleoprotein complex (lanes 11–15). These data are consistent with HapR having a higher affinity for CRP-P*murQP* than P*murQP* alone. Hence, HapR and CRP bind the *murQP* regulatory region co-operatively. A mundane explanation is that increased molecular crowding, upon CRP addition, increases the effective concentration of HapR.

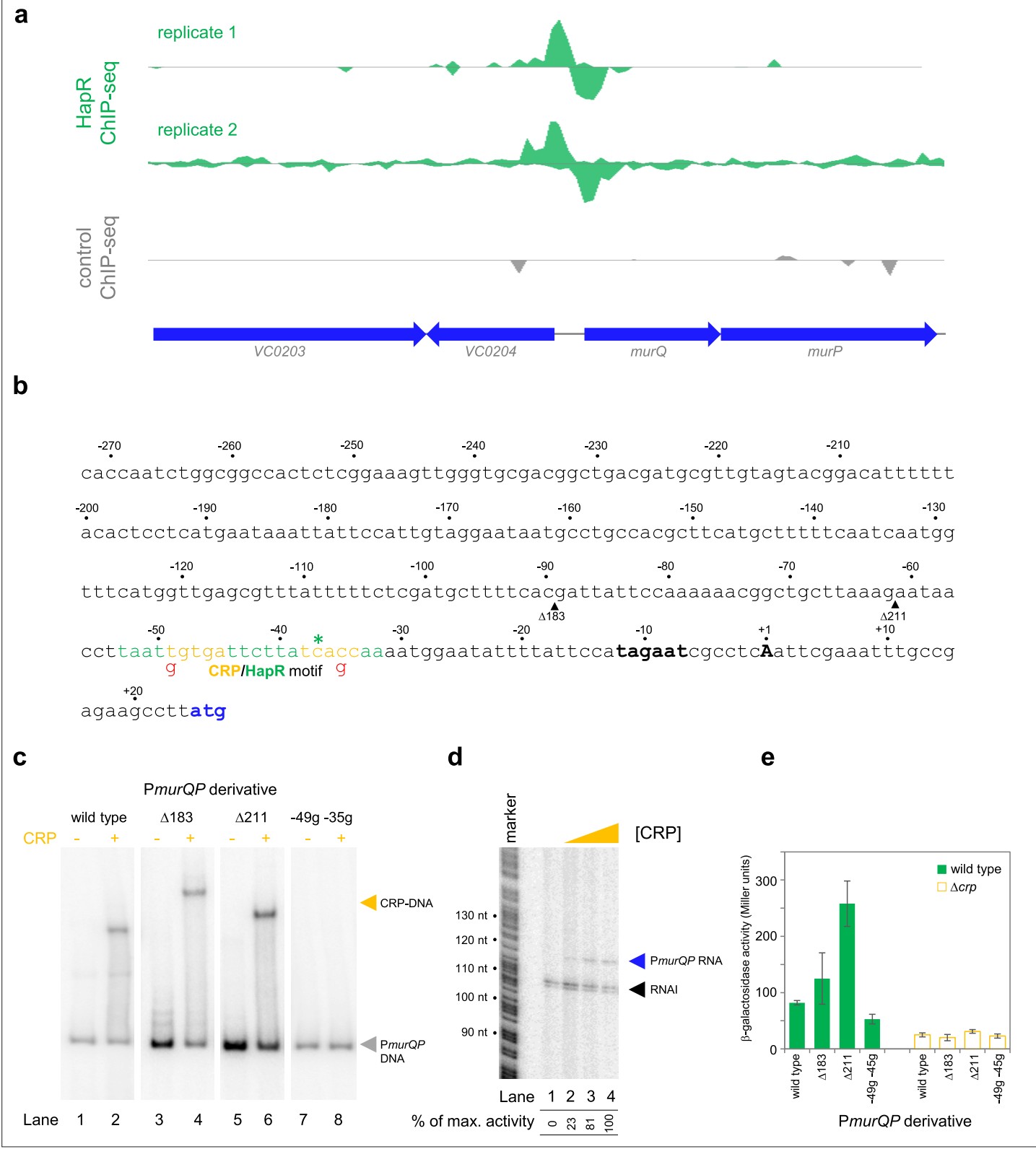

**Figure 3.** Transcription from the *murQP* promoter requires CRP in vivo and in vitro. (**a**) HapR binding to the *murQP* regulatory region. Genes are shown as block arrows. ChIP-seq coverage plots are shown for individual experimental replicates. Signals above or below the horizontal line correspond to reads mapping to the top or bottom strand respectively. (**b**) DNA sequence of the intergenic region upstream of *murQP*. For clarity, numbering is with respect to the *murQP* transcription start site (TSS,+1). The TSS and promoter –10 element are in bold. The *murQ* start codon is in blue. The

*Figure 3 continued on next page*

*Figure 3 continued*

HapR binding site, predicted by MEME analysis of our ChIP-seq data for HapR, is in green. A potential CRP site is embedded within the HapR binding sequence (orange). Sequences in red indicate point mutations used in this work. Triangles show sites of truncation. (**c**) **Binding of CRP to the** *murQP* **regulatory region and derivatives**. Electrophoretic mobility shift assays showing migration of the *murQP* regulatory region, or indicated derivatives, with or without 0.1 µM CRP. The DNA fragment used is shown above each pair of lanes and correspond to the truncations or point mutations indicated in panel b. (**d**) **The** *murQP* **promoter is activated by CRP in vitro**. The gel image shows the result of an in vitro transcription assay. The DNA template was plasmid pSR carrying the *murQP* regulatory region. Experiments were done with 0.4 µM RNA polymerase with or without 0.125, 0.25, or 0.5 µM CRP. The RNAI transcript is plasmid-derived and acts as an internal control. (**e**) **The** *murQP* **promoter is activated by CRP in vivo**. The bar chart shows results of β-galactosidase activity assays. Cell lysates were obtained from wild type *V. cholerae* E7946 (solid green) or the Δ*crp* derivative, transformed with pRW50T derivatives containing the indicated promoter derivatives fused to *lacZ*. Standard deviation is shown for three independent biological replicates. Cells were grown in LB-Lennox medium at 37 °C to an $OD_{650}$ of ~1.1.

The online version of this article includes the following source data for figure 3:

**Source data 1.** Gel image TIFF file, *Figure 3c*.

**Source data 2.** Gel image TIFF file, *Figure 3c*.

**Source data 3.** Gel image TIFF file, *Figure 3d*.

To exclude this possibility, we did two further sets of EMSA experiments. In the first set of assays, CRP was added at a lower concentration. Thus, some DNA remained unbound (*Figure 4c*, lanes 1 and 2). Hence, when added to such reactions, HapR could bind either the free DNA or the CRP-DNA complex. Consistent with HapR preferentially binding the latter, all of the CRP-DNA complex was super shifted upon HapR addition. Conversely, the free DNA remained unbound (compare lanes 2 and 4). In equivalent experiments, with point mutations −49 g and −35 g in the CRP site, neither CRP or HapR were able to bind the DNA (lanes 5–8). In a second set of tests, we used the *hapR* regulatory DNA that binds HapR but not CRP. If CRP addition increased the effective concentration of HapR, this should result in much tighter HapR binding to the *hapR* promoter. However, this was not the case (*Figure 4—figure supplement 1*). Taken together, our data are consistent with CRP and HapR co-operatively binding the same DNA locus at the *murQP* promoter region.

## HapR represses CRP dependent transcription from the murQP promoter in vivo and in vitro

Recall that, in the absence of CRP, P*murQP* is inactive in vitro (*Figures 2b and 3d*). Furthermore, the promoter is subject to repression by HapR in vivo (*Figure 2a*). An explanation consistent with both observations is that HapR directly counteracts CRP mediated activation. To test this, we used in vitro transcription assays (*Figure 4d*). As expected, addition of CRP activated *murQP* transcription (lanes 1–4) and this was blocked by addition of HapR (lanes 5–8). We also repeated our prior *lacZ* fusion experiments, using the Δ211 P*murQP* derivative, and *V. cholerae* E7946 lacking *crp* and/or *hapR*. The result is shown in *Figure 4e*. Deletion of *hapR* caused increased transcription from P*murQP* only when CRP was present. Hence, HapR also represses CRP dependent *murQP* transcription in vivo.

## Binding sites for CRP and HapR overlap in a specific configuration genome-wide

Both CRP and HapR bind the same DNA region upstream of *murPQ*. This suggests similar nucleic acid sequences are recognised by each factor. *Figure 5a* shows an alignment of DNA logos, derived from CRP (*Manneh-Roussel et al., 2018*) (top) and HapR (bottom) ChIP-seq targets. The two motifs have features in common that align best when the logo centres are offset by 1 base pair. This is consistent with the arrangement of binding sites upstream of *murPQ* (*Figure 3b*). To understand the importance of this configuration, we first took a bioinformatic approach. The DNA sequences logos shown in *Figure 5a* were used to create position weight matrices (PWMs) describing either the CRP or HapR binding site. We then searched the *V. cholerae* genome, using each PWM, and calculated the distance between identified CRP and HapR sites. The data for all sites within 100 bp of each other is shown in *Figure 5b* (top panel). In all cases, the CRP and HapR targets were offset by 1 bp. We then repeated the analysis after randomising the *V. cholerae* genome sequence (bottom panel). The number of overlapping targets was reduced 7-fold. An equivalent analysis of the *V. harveyi* genome produced similar results (*Figure 5—figure supplement 1*). Hence, sites for CRP and HapR have a

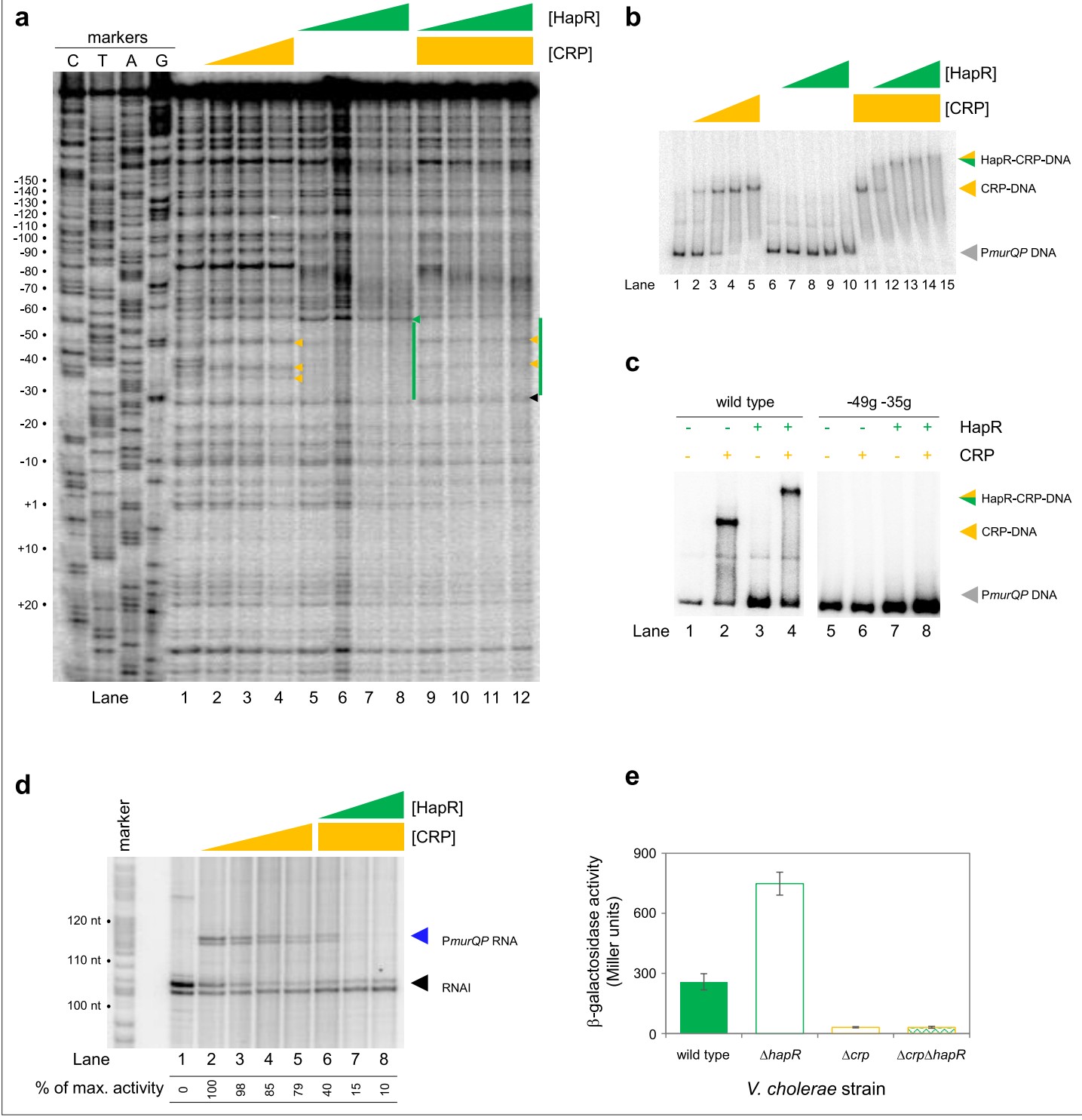

**Figure 4.** HapR and CRP co-operatively bind the same section of *murQP* regulatory DNA. (**a**) Binding locations of HapR and CRP upstream of *murQP*. The gel shows the result of DNase I footprinting experiment. The gel is calibrated with Sanger sequencing reactions. The pattern of DNase I cleavage in the absence of any proteins is in lane 1. Protection of DNA from DNase I cleavage in the presence of 0.11, 0.23 or 0.45 µM CRP is shown in lanes 2–4. Sites of DNAse I hypersensitivity due to CRP binding are indicated by orange triangles. Protection from DNase I cleavage in the presence of 0.5, 1.0, 2.0 or 3.0 µM HapR is shown in lanes 5–8. Protection from DNase I cleavage, dependent on HapR, is shown by a green bar. A DNAse I hypersensitive band, unique to reactions with HapR, is shown by a green triangle. In the presence of 0.45 µM CRP, increasing concentrations of HapR result in a different DNAse I cleavage pattern, including the appearance of a different site of hypersensitivity (black triangle). (**b**) Binding of HapR and CRP upstream of *murQP* is co-operative. Electrophoretic mobility shift assays showing migration of the *murQP* regulatory region with different combinations of CRP

*Figure 4 continued on next page*

*Figure 4 continued*

(0.025, 0.05, 0.1 or 0.2 μM) and HapR (0.5, 1.0, 2.0, 3.0 or 4.0 μM). For incubations with both factors, the same range of HapR concentrations was used with 0.2 μM CRP. (**c**) Co-operative binding of CRP requires the shared HapR and CRP binding site. Results of an electrophoretic mobility shift assay, using the wild type *murQP* regulatory region or a derivative with two point mutations in the shared recognition sequence, for HapR (4.0 μM) and CRP (0.1 μM). Positions of mutations are shown in *Figure 3b*. (**d**) HapR blocks CRP mediated activation of the *murQP* promoter in vitro. The gel image shows the result of an in vitro transcription assay. The DNA template was plasmid pSR carrying the *murQP* regulatory region. Experiments were done with 0.4 μM RNA polymerase, with or without 0.05, 0.1, 0.2 or 0.5 μM CRP 0.5, 1.0, 2.0 or 3.0 μM HapR, as indicated. The RNAI transcript is plasmid-derived and acts as an internal control. (**e**) HapR represses CRP mediated activation of the *murQP* promoter in vivo. β-galactosidase activity was measured in cell lysates taken from *Vibrio cholerae* E7946 (solid green bars), Δ*hapR* derivative (open green bars), Δ*crp* variant (open orange bars), or cells lacking both factors (orange outline with green patterned fill). Standard deviation is shown for three independent biological replicates. Cells were grown in LB-Lennox medium to an OD$_{650}$ of ~1.0 at 37 °C.

The online version of this article includes the following source data and figure supplement(s) for figure 4:

**Source data 1.** Gel image TIFF file, *Figure 4a*.

**Source data 2.** Gel image TIFF file, *Figure 4b*.

**Source data 3.** Gel image TIFF file, *Figure 4c*.

**Source data 4.** Gel image TIFF file, *Figure 4c*.

**Source data 5.** Gel image TIFF file, *Figure 4d*.

**Figure supplement 1.** Binding of HapR to the *hapR* promoter region in the presence and absence of CRP.

**Figure supplement 1—source data 1.** Gel image TIFF file.

propensity to coincide in a specific configuration. That such sites are found more frequently in native genome sequences, compared to those first randomised, suggests selection during genome evolution. We next sought to understand how this arrangement might permit simultaneous and co-operative binding of CRP and HapR.

## A structural model of the DNA-CRP-HapR ternary complex

To understand organisation of the DNA-CRP-HapR ternary complex we used structural modelling. The *V. cholerae* CRP protein is 96% identical to the equivalent factor in *Escherichia coli*. Similarly, the *Staphylococcus aureus* factor QacR is 50% similar to *V. cholerae* HapR. Previously, structural biology tools were used to investigate *E. coli* CRP, and *S. aureus* QacR, bound with their cognate DNA targets. We used this information to build a model for the DNA-CRP-HapR ternary complex. Importantly, we ensured that the CRP and HapR binding centres were offset by 1 bp. When aligned in this way, CRP and HapR recognise the same section of DNA via different surfaces of the double helix. We examined the model in the context of our DNAse I footprinting data. Recall that CRP binding upstream of *murPQ* induces three sites of DNAse I hypersensitivity (*Figure 4a*). These correspond to positions −47, −38, and −34 with respect to the *murQP* TSS. *Figure 5—figure supplement 2* shows these positions highlighted in the context of our model. In the presence of CRP alone, all sites are surface exposed but position −34 is partially occluded by CRP (*Figure 5—figure supplement 2a*). This likely explains why positions −47 and −38 are more readily cleaved by DNAse I (*Figure 4a*). With both CRP and HapR, position −34 was completely protected from DNAse I attack (*Figure 4a*). Consistent with the footprinting data, our model indicates that position −34 is almost completely hidden upon binding of HapR (*Figure 5—figure supplement 2b*). Conversely, access to positions −47 and −38 is not altered (Compare *Figure 5—figure supplement 2a b*).

## Co-operative binding with HapR requires CRP residue E55

Co-operative DNA binding by transcription factors can result from their direct interaction (*Wade et al., 2001*; *Kallipolitis et al., 1997*; *Meibom et al., 2000*). In our model, a negatively charged surface of CRP (including residue E55) is in close proximity to positively charged HapR residue R123 (*Figure 5c*). In initial experiments, we mutated both protein surfaces to remove the charged side chain, or replace the residue with an oppositely charged amino acid. We then investigated consequences for HapR and CRP binding individually at P*murQP* using EMSAs (*Figure 5—figure supplement 3*). Whilst the CRP derivatives were able to bind the *murQP* regulatory region normally, HapR variants were completely defective. This is likely because R123 sits at the HapR dimerisation interface. Hence, we focused on understanding the contribution of CRP sidechain E55 to co-operative DNA binding by HapR and CRP

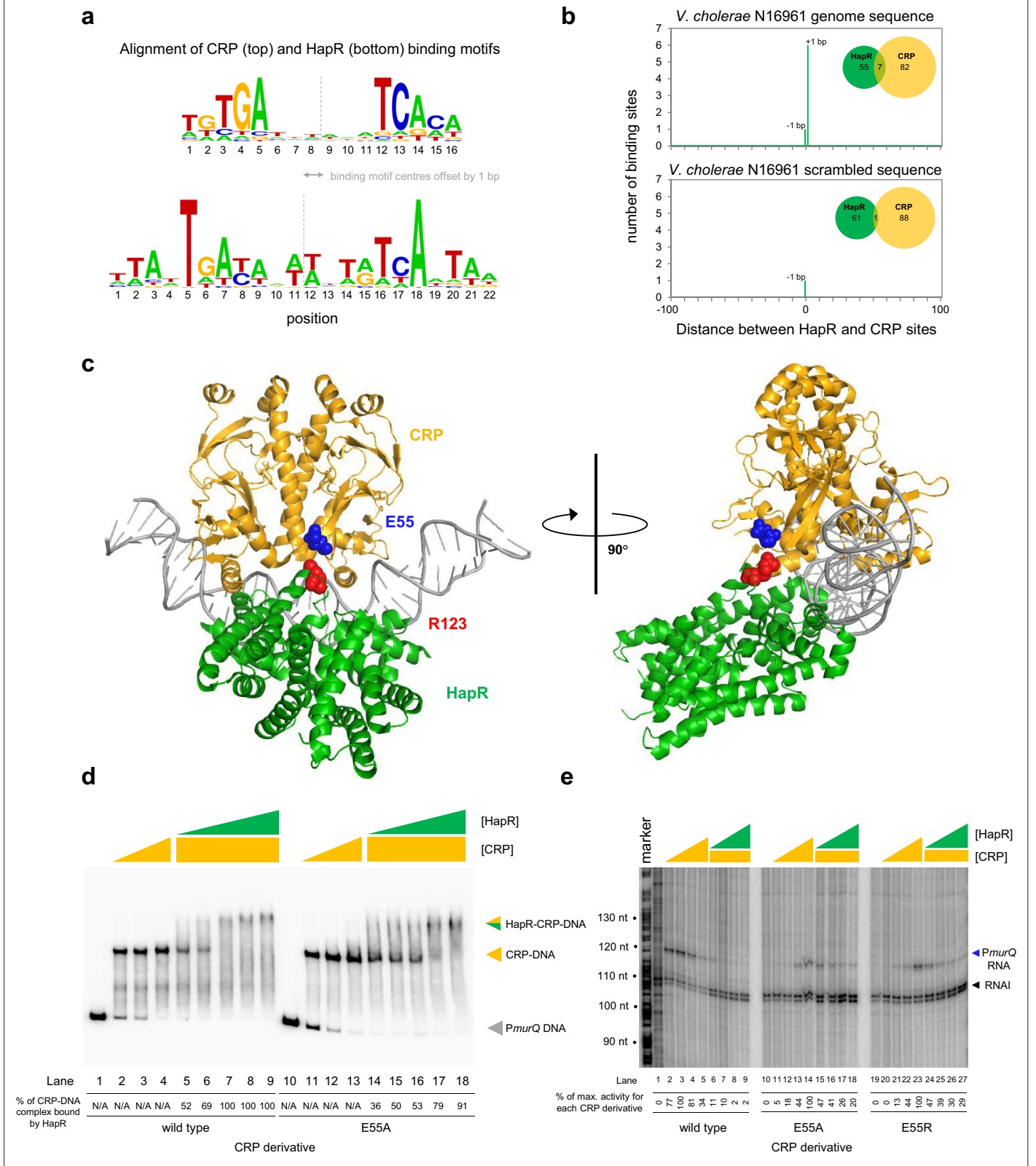

**Figure 5.** HapR contacts Activation Region 3 of CRP. (**a**) **Binding sites for CRP and HapR are optimally aligned when offset by one base pair**. The panel shows DNA sequences logos generated by aligning binding sites identified by ChIP-seq analysis for CRP (top) and HapR (bottom). The centre of each motif is indicated by a dashed line. (**b**) **Global overlap of CRP and HapR binding sites**. A position weight matrix (PWM), corresponding to each DNA sequence logo shown in panel a, was created. The PWMs were used to search the *V. cholerae* genome sequence using FIMO. Distances between the

*Figure 5 continued*

identified CRP and HapR sites were calculated. Proximal sites were always overlapping and offset by one base pair (top panel). Overlap was greatly reduced when the analysis was applied to a randomised version of the same genome sequence (bottom panel). (**c**) **Model of the DNA-CRP-HapR complex.** The model was generated using PDB submissions 6pb6 (*E. coli* CRP in complex with a class II CRP dependent promoter) and 1jt0 (*S. aureus* QacR bound to its DNA target). Note that QacR is closely related to *V. cholerae* HapR. The structures were aligned so that the CRP and HapR binding centres were offset by one base pair. Residue E55 of CRP (blue) is within Activating Region 3 of CRP that can interact with the RNA polymerase sigma subunit at class II promoters. HapR residue R123 (red) participates in HapR dimerisation and is proximal to E55 of CRP. (**d**) **Side chain E55 of CRP is required for stability of the DNA-CRP-HapR complex.** Electrophoretic mobility shift assays showing migration of the *murQP* regulatory region with different combinations of CRP or CRP$^{E55A}$ (0.15, 0.3, or 0.6 µM) and HapR (0.083, 0.125, 0.166, 0.208, or 0.25 µM). (**e**) **HapR cannot repress transcription activated by CRP$^{E55A}$.** Result of an in vitro transcription assay. The DNA template was plasmid pSR carrying the *murQP* regulatory region. Experiments were done with 0.4 µM RNA polymerase, with or without 0.05, 0.1, 0.2, or 0.5 µM CRP or CRP$^{E55A}$ and 0.5, 1.0, 2.0, or 3.0 µM HapR, in the presence of 0.2 µM CRP, as indicated. The RNAI transcript is plasmid-derived and acts as an internal control.

The online version of this article includes the following source data and figure supplement(s) for figure 5:

**Source data 1.** Gel image TIFF file, *Figure 5c*.

**Source data 2.** Gel image TIFF file, *Figure 5e*.

**Figure supplement 1.** Global overlap of CRP and HapR binding sites in *Vibrio harveyi*.

**Figure supplement 2.** Models of the DNA-CRP and DNA-CRP-HapR complexes.

**Figure supplement 3.** Binding of CRP and HapR derivatives to P*murQP*.

**Figure supplement 3—source data 1.** Gel image TIFF file.

**Figure supplement 3—source data 2.** Gel image TIFF file.

**Figure supplement 4.** Co-operative DNA binding of HapR and CRP is common.

**Figure supplement 4—source data 1.** Gel image TIFF file.

**Figure supplement 4—source data 2.** Gel image TIFF file.

**Figure supplement 4—source data 3.** Gel image TIFF file.

**Figure supplement 4—source data 4.** Gel image TIFF file.

---

using EMSAs. The results are shown in *Figure 5d*. Both wild type CRP, and CRP$^{E55A}$, were able to bind the *murQP* regulatory region similarly (lanes 1–4 and 10–13). As expected, HapR bound tightly to the wild type CRP:DNA complex (lanes 5–9). Conversely, HapR had a lower affinity for DNA in complex with CRP$^{E55A}$ (lanes 14–18). This suggests that the E55A mutation in CRP destabilises the interaction with HapR.

### Repression of P*murQP* by HapR requires CRP residue E55

Residue E55 locates to a negatively charged surface of CRP called Activating Region 3 (AR3). This determinant aids recruitment of RNA polymerase when CRP binds close to the promoter –35 element (*Rhodius and Busby, 2000*). Hence, AR3 is likely to be important for activation of P*murQP* (*Figure 3b*). We inferred that CRP lacking E55 should activate P*murQP* less efficiently but be less sensitive to negative effects of HapR. To test these predictions, we used in vitro transcription assays. The results for CRP, CRP$^{E55A}$ and CRP$^{E55R}$ are shown in *Figure 5e*. All CRP derivatives were able to activate transcription from P*murQP*. However, consistent with an important role for AR3, the ability of the CRP$^{E55A}$ and CRP$^{E55R}$ to activate transcription was impaired (compare lanes 1–5, 10–14, and 19–23). Crucially, whilst HapR reduced transcription dependent on wild type CRP by 50-fold (compare lane 4 with lanes 6–9) only a 2-fold effect of HapR was observed with CRP$^{E55A}$ (compare lane 13 with lanes 15–18). In the presence of CRP$^{E55R}$, HapR was even less effective (compare lane 22 with lanes 24–27).

### High cell density locked *V. cholerae* are defective for growth on MurNAc

Phosphorylated LuxO activates expression of the Qrr sRNAs that inhibit *hapR* expression at low cell density (*Figure 1a*). Consequently, deletion of *luxO* causes constitutive expression of HapR. Thus, Δ*luxO V. cholerae* are 'locked' in a high cell density state (*Waters et al., 2008*). Our model predicts that such strains will be defective for growth using MurNAc as the sole carbon source, as this requires expression of *murQP* that is repressed by HapR. Furthermore, any such defect should be relieved upon deletion of *hapR*. To test this, we constructed strains lacking different combinations of *luxO* and

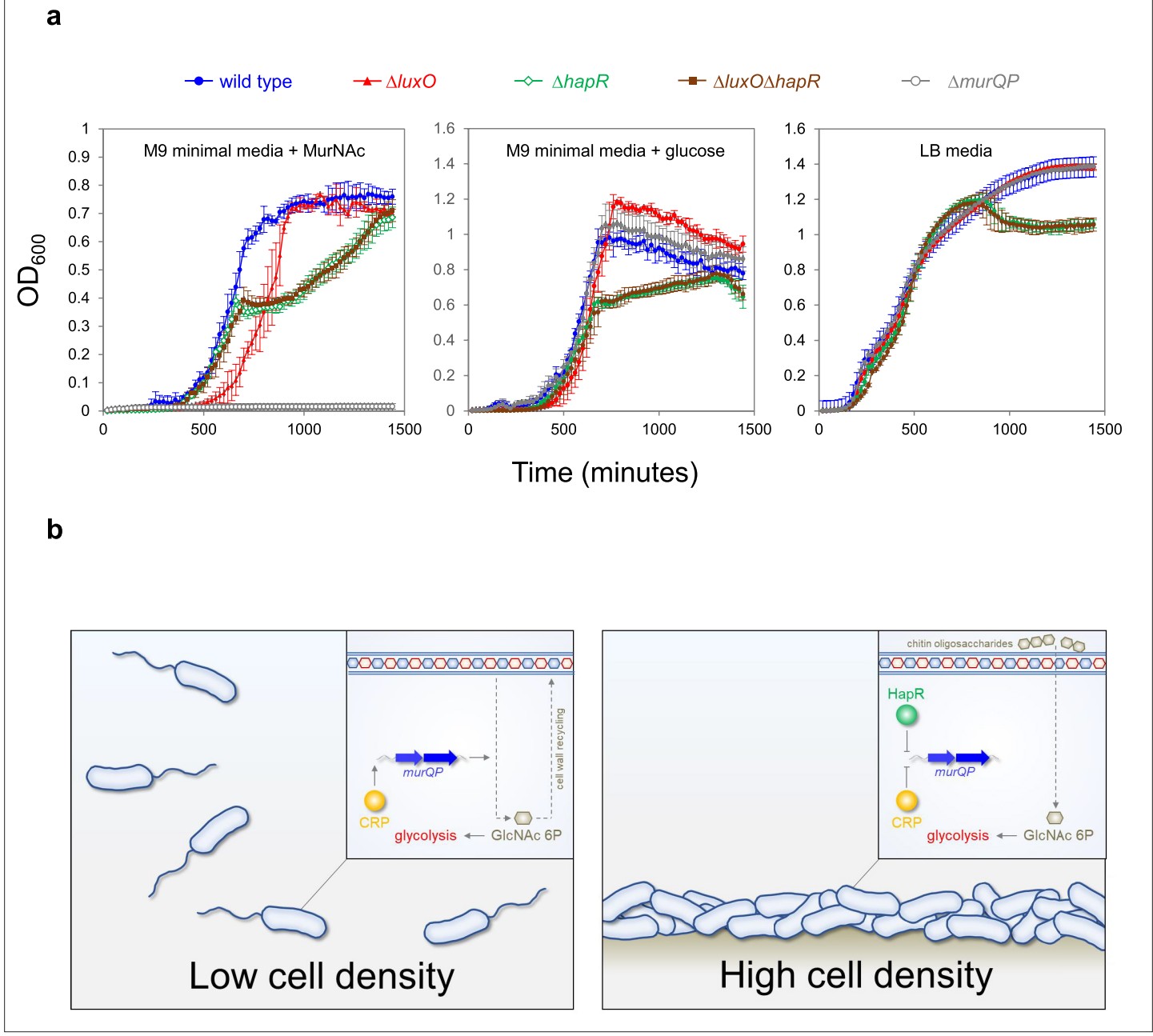

**Figure 6.** Control of *murQP* expression by CRP and HapR at low and high cell density. (**a**) *V. cholerae* locked at high cell density are defective for growth using MurNAc as the sole carbon source. Each panel illustrates the optical density of *V. cholerae* cultures at different timepoints after inoculation. Cells were grown at 32 °C in M9 minimal media supplemented with the indicated carbon source (0.25 % *w/v*) or LB-Miller medium. Cells lacking *luxO*, but not *luxO* and *hapR*, mimic the high cell density state. Error bars show standard deviation from three separate experimental replicates. (**b**) Model for coordination of MurNAc catabolism by CRP and HapR. In low *V. cholerae* population density conditions (left panel) cell division necessitates cell wall turnover. Expression of MurQP facilitates cell wall recycling and conversion of MurNAc to GlcNAc 6 P for glycolysis (insert). At high cell density conditions (right panel) *V. cholerae* form biofilms on chitinous surfaces. Reduced cell division, and the availability of chitin derived GlcNAc 6 P, reduces the need for MurQP.

*hapR*. We also tested a *V. cholerae* derivative lacking *murP* (**Hayes et al., 2017**). **Figure 6a** illustrates growth in M9 minimal media, supplemented with MurNAc or glucose, and in LB-Miller medium. As expected, cells lacking *murP* could not grow when MurNAc was the only carbon source but were not defective in other conditions (compare grey data points in each panel). Cells lacking *hapR*, alone or in combination with *luxO*, had a similar growth defect in all conditions. Strikingly, the *luxO* mutant

(high cell density locked), exhibited a growth defect only when MurNAc was the sole carbon source (compare red data points). Specifically, these cells exhibited an extended lag phase in MurNAc. This extended lag phase was not apparent when both *luxO* and *hapR* were deleted, consistent with the effect of *luxO* being mediated by HapR-dependent repression of *murQP*.

## Co-operative interactions between HapR and CRP are commonplace

In a final set of experiments, we turned our attention to other sites shared by CRP [prior work (*Manneh-Roussel et al., 2018*)] and HapR (*Table 1*). We selected 5 such targets and examined binding of CRP and HapR using EMSAs. At 1 target, adjacent to *VCA0218*, binding was not co-operative and free DNA remained when both proteins were present (*Figure 5—figure supplement 4*). For 4 of the targets, we detected co-operative binding of CRP and HapR, reminiscent of our experiments with P*murQP* DNA (compare *Figure 5—figure supplement 4* and *Figure 4*). At these loci (adjacent to *VC0102, VC1851, VCA0663,* and *VCA0691*) either HapR or CRP bound poorly to DNA in the absence of the other protein. However, when both factors were added together, all DNA shifted into a distinct low mobility complex. We conclude that co-operative binding of HapR and CRP to shared targets is common.

## Discussion

Previously, two studies have mapped DNA binding by HapR homologs in *Vibrio* species. For *V. harveyi*, van Kessel and co-workers used ChIP-seq to identify 105 LuxR binding targets (*van Kessel et al., 2013b*). At 77 of these sites, LuxR repressed transcription. Using ChIP-seq and global DNAse I footprinting, Zhang et al. found 76 LuxR bound regions in *Vibrio alginolyticus* (*Zhang et al., 2021*). Regulatory effects were evident for 37 targeted genes, with 22 cases of LuxR mediated repression. In the present study, we identified 32 HapR bound sections of the *V. cholerae* genome. Consistent with prior work, repression of target genes was the most common regulatory outcome. Furthermore, the DNA binding consensus derived here for HapR is almost identical to motifs for LuxR binding in *V. harveyi* and *V. alginolyticus.* Contrastingly, Tsou and colleagues used bioinformatic tools to predict HapR binding in *V. cholerae* (*Tsou et al., 2009*). Two different HapR binding motifs were proposed. Both partially match the HapR target sequence proposed here. Most likely, the analysis of Tsou et al. was hampered by a paucity of targets from which a full consensus could be derived. We note that our list of 32 HapR targets does not include all known targets. However, on inspection, whilst insufficient to pass our stringent selection criteria, weaker signals for HapR are evident at many such locations (*Figure 1—figure supplement 2* and *Supplementary file 1*). In particular, we note evidence for binding of HapR upstream of *hapA*, which has previously been only inferred (*Figure 1—figure supplement 2b*). We note that previous work relied on computational predictions and in vitro DNA binding assays to identify potential HapR targets. That not all such targets are bound in vivo, in the single growth condition tested here, is to be expected.

Recognition of shared DNA targets provides a simple mechanism for integration of quorum sensing signals, relayed by HapR, and cAMP fluctuations, communicated by CRP. In the example presented here, HapR acts to prevent transcription activation by co-binding the same DNA target with CRP (*Figure 4*). Hence, at P*murQP*, the function of CRP switches from that of an activator to a co-repressor with HapR (*Figure 6b*). This regulatory strategy is a logical consequence of *V. cholerae* forming biofilms on chitinous surfaces. At low cell density, rapidly dividing cells must continually remodel their cell wall. In these conditions, HapR is not expressed. Thus, MurQ and MurP are produced and can convert cell wall derived MurNAc to GlcNAc-6P. Conversely, in high cell density scenarios, usually involving adherence to chitin, cells divide infrequently, and remodelling of the cell wall is reduced. In addition, GlcNAc-6P can be derived readily from chitin oligosaccharides. Hence, cells locked in the high cell density state are defective for growth when supplied with MurNAc as the sole carbon source (*Figure 6a*). We suggest that HapR and CRP are likely to coordinate the expression of other metabolic enzymes in a similar way. Interestingly, AphA, another quorum sensing responsive regulator, also acts alongside CRP at many *V. cholerae* promoters (*Haycocks et al., 2019*). Indeed, AphA and CRP binding sites can overlap but this results in competition between the factors (*Haycocks et al., 2019*). Together with results presented here, these observations highlight close integration of quorum sensing with gene control by cAMP in *V. cholerae*.

## Materials and methods

### Strains, plasmids and oligonucleotides

Strains, plasmids and oligonucleotides used in this study are listed in *Supplementary file 2*. All *V. cholerae* strains are derivatives of E7946 (*Levine et al., 1982*). Chromosomal deletions were made using the pKAS32 suicide plasmid for allelic exchange (*Skorupski and Taylor, 1996*; *Dalia et al., 2014*) or via splicing-by-overlap-extension PCR and chitin-induced natural transformation (*Dalia, 2018*). The *E. coli* strain JCB387 was used for routine cloning (*Page et al., 1990*). Plasmids were transferred into *V. cholerae* by either conjugation or transformation as described previously (*Manneh-Roussel et al., 2018*; *Haycocks et al., 2019*).

### ChIP-seq and bioinformatics

Chromatin immunoprecipitation was done as in prior work (*Haycocks et al., 2019*) using strain E7946, carrying plasmid pAMCF-*luxO* or pAMNF-*hapR*, and anti-FLAG antibodies. In both cases, control experiments were done using the equivalent plasmid with no gene insert. Note that both plasmids drive low level constitutive expression of 3xFLAG transcription factor derivatives (*Sharma et al., 2017*). Lysates were prepared from LB-Lennox medium cultures, incubated with shaking at 37 °C to an $OD_{650}$ of ~1.1. Following sonication, the protein-DNA complexes were immunoprecipitated with an anti-FLAG antibody (Sigma) and Protein A sepharose beads. Immunoprecipitated DNA was blunt-ended, A- tailed, and ligated to barcoded adaptors before elution and de-crosslinking. ChIP-seq libraries were then amplified by PCR and purified. Library quality was assessed using an Agilent Tapestation 4200 instrument and quantity determined by qPCR using an NEBnext library quantification kit (NEB). Libraries were sequenced as described previously (*Sharma et al., 2017*) and reads are available from ArrayExpress using accession code E-MTAB-11906. Single-end reads, from two independent ChIP-seq experiments for each strain, were mapped to the reference *V. cholerae* N16961 genome (chromosome I: NC_002505.1 and chromosome II: NC_002506.1) with Bowtie 2 (*Langmead and Salzberg, 2012*). The read depth at each position of the genome was determined for each BAM file using multibamsummary. Each binding profile was then normalised to an average genome-wide read depth of 1 read per base. Following normalisation, the average read depth per base for each pair of replicates was calculated. The resulting files were used to generate the circular plots in *Figure 1* using DNAplotter (*Carver et al., 2009*). For peak selection, the files were viewed as graphs using the Artemis genome browser (*Carver et al., 2012*). After visually identifying an appropriate cut-off, peaks were selected using the 'create features from graph' tool. Note that our cut-off was selected to identify only completely unambiguous binding peaks. Hence, weak or less reproducible binding signals, even if representing known targets, were excluded (see Discussion for further details). For HapR, the window size, minimum feature size, and cut-off value were 100, 100, and 10, respectively. For LuxO, the equivalent values were 100, 100, and 4. The mid-point of features selected in this way was set as the peak centre. In each case, 300 bp of sequence from the peak centre was selected and the combined set of such sequences for each factor were analysed using MEME to generate DNA sequence logos (*Bailey et al., 2009*).

### β-galactosidase assays

Promoter DNA was fused to *lacZ* in plasmid pRW50T that can be transferred from *E. coli* to *V. cholerae* by conjugation (*Manneh-Roussel et al., 2018*). Assays of β-galactosidase activity were done according to the Miller method (*Miller, 1972*). Bacterial cultures were grown at 37 °C with shaking in LB-Lennox medium, supplemented with appropriate antibiotics, to an $OD_{650}$ of ~1.1. Values shown are the mean of three independent experiments and error bars show the standard deviation.

### Proteins

We purified *V. cholerae* CRP and RNA polymerase as described previously (*Manneh-Roussel et al., 2018*; *Haycocks et al., 2019*). To generate HapR, *E. coli* T7 Express cells were transformed with plasmid pHis-tev-HapR, or derivatives, which encodes HapR with a $His_6$ tag and intervening site for the tobacco etch virus protease protease. Transformants were cultured in 40 ml LB-Lennox medium overnight, then sub-cultured in 1 L of fresh broth, with shaking at 37 °C. When sub-cultures reached mid-log phase they were supplemented with 400 mM IPTG for 3 hr. Cells were then collected by centrifugation, resuspended in 40 ml of buffer 1 (40 ml 25 mM Tris-HCl pH 7.5, 1 mM EDTA and 1 M

NaCl) and lysed by sonication. Inclusion bodies, recovered by centrifugation, were resuspended with 40 ml of buffer 2 (25 mM Tris-HCl pH 8.5 and 4 M urea) before the remaining solid material was again recovered and then solubilised using 40 ml of buffer 3 (25 mM Tris-HCl pH 8.5 and 6 M guanidine hydrochloride). Cleared supernatant was applied to a HisTrap HP column (GE healthcare) equilibrated with buffer A (25 mM Tris-HCl pH 8.5 and 1 M NaCl). To elute $His_6$-HapR, a gradient of buffer B (25 mM Tris-HCl pH 8.5, 1 M NaCl and 1 M imidazole) was used. Fractions containing $His_6$-HapR were pooled and the protein was transferred into buffer X (50 mM HEPES, 1 M NaCl, 1 mM DTT, 5 mM EDTA and 0.1 mM Triton X-100) by dialysis. Finally, we used Vivaspin ultrafiltration columns to reduce sample volume. The concentration of $His_6$-HapR was then determined.

### in vitro *transcription assays*

Experiments were done using our prior approach (*Haycocks et al., 2019*). Plasmid templates were isolated from *E. coli* using Qiagen Maxiprep kits. Each in vitro transcription assay contained 16 µg/ml DNA template in 40 mM Tris pH 7.9, 5 mM MgCl2, 500 µM DTT, 50 mM KCl, 100 µg/ml BSA, 200 µM ATP/GTP/CTP, 10 µM UTP and 5 µCi α-P32-UTP. Purified HapR and CRP were added at the indicated concentrations prior to the reaction start point. In experiments where CRP was used, the protein was incubated with cAMP 37 °C prior to addition. Transcription was instigated by addition of RNA polymerase holoenzyme prepared in advance by incubation of the core enzyme with a 4-fold excess of $\sigma^{70}$ for 15 min at room temperature. After 10 min incubation at 37 °C, reactions were stopped by the addition of an equal volume of formamide containing stop buffer. Reactions were resolved on an 8% (*w/v*) denaturing polyacrylamide gel, exposed on a Bio-Rad phosphor screen then visualised on a Bio-Rad Personal Molecular Imager. The quantify transcript levels, we measured the intensity of bands corresponding to RNAI and the RNA of interest using Quantity One software. After subtracting background lane intensity, we calculated the RNA of interest to RNAI ratio. The maximum ratio was set to 100% activity with other ratios shown a percentage of this maximum. Experiments were repeated at least twice with similar results.

### Electrophoretic mobility shift assays and DNAse I footprinting

Promoter DNA fragments were excised from plasmid pSR and end-labelled with γ32-ATP using T4 PNK (NEB). EMSAs and DNase I footprints were done as previously described (*Haycocks et al., 2019*). Full gel images are shown in *Figure 2—source data 7*. Experiments were repeated at least twice with similar results.

### Structural modelling

The model of the ternary DNA-CRP-HapR complex was generated in PyMOL by aligning PDB depositions 1jt0 (QacR-DNA complex) and 6pb6 (CRP-DNA complex). Alignments were done manually and guided by the relative two-fold centres of symmetry for each complex. Each structure was positioned so that their DNA base pairs overlapped and binding centres were offset by 1 base pair. The Mutagenesis function of PyMOL was used to replace QacR sidechain K107, equivalent to HapR R123 (*De Silva et al., 2007*), with an arginine residue. The double helix of the QacR DNA complex is hidden in the final model.

### Materials availability statement

Strains, plasmids and oligonucleotides are available on request.

## Acknowledgements

This work was funded by BBSRC project grant BB/N005961/1 awarded to DCG, a BBSRC MIBTP studentship (BB/M01116X/1, project reference 1898542) awarded to LMW, and grant R35GM128674 from the National Institutes of Health (to ABD). We thank Kai Papenfort, Jenny Ritchie and Joseph Wade, for commenting on the manuscript prior to submission, and Melanie Blokesch for helpful discussions.

# Additional information

## Funding

| Funder | Grant reference number | Author |
|---|---|---|
| Biotechnology and Biological Sciences Research Council | BB/N005961/1 | David C Grainger |
| Biotechnology and Biological Sciences Research Council | BB/M01116X/1 | Lucas M Walker |
| National Institutes of Health | R35GM128674 | Ankur B Dalia |
| Biotechnology and Biological Sciences Research Council | project reference 1898542 | Lucas M Walker |

The funders had no role in study design, data collection and interpretation, or the decision to submit the work for publication.

## Author contributions

Lucas M Walker, Formal analysis, Investigation, Methodology, Writing – review and editing; James RJ Haycocks, Formal analysis, Supervision, Investigation, Methodology, Writing – review and editing; Julia C Van Kessel, Writing – review and editing; Triana N Dalia, Investigation, Methodology; Ankur B Dalia, Investigation, Methodology, Writing – review and editing; David C Grainger, Conceptualization, Data curation, Supervision, Funding acquisition, Investigation, Writing - original draft

## Author ORCIDs

Julia C Van Kessel ⓘ http://orcid.org/0000-0002-1612-2403
David C Grainger ⓘ http://orcid.org/0000-0003-3375-5154

Reviewer #1 (Public Review): https://doi.org/10.7554/eLife.86699.3.sa1
Reviewer #2 (Public Review): https://doi.org/10.7554/eLife.86699.3.sa2
Author Response: https://doi.org/10.7554/eLife.86699.3.sa3

# Additional files

## Supplementary files

• Supplementary file 1. HapR binding targets identified by previous studies.
• Supplementary file 2. Strains, plasmids, and oligonucleotides.
• MDAR checklist

## Data availability

Sequencing reads are available from ArrayExpress using accession code E-MTAB-11906.

The following dataset was generated:

| Author(s) | Year | Dataset title | Dataset URL | Database and Identifier |
|---|---|---|---|---|
| Grainger D | 2023 | ChIP-seq analysis of genome-wide DNA binding by transcription factors HapR and LuxO in *Vibrio cholerae* | https://www.ebi.ac.uk/biostudies/arrayexpress/studies/E-MTAB-11906 | ArrayExpress, E-MTAB-11906 |

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
