## [Editor Report · eLife assessment]

This paper provides **valuable** new information on the mechanisms by which *Vibrio cholerae* integrates and responds to environmental signals. The strength of the evidence provided in support of the conclusions made and the model proposed is **solid**. The revision resolved many of the issues raised by the reviewers and improved the manuscript. The work is relevant for a broad audience of microbiologists interested in the mechanisms by which bacteria sense their environment.

---

## [Referee Report · Reviewer #1 (Public Review)]

Original review:

This manuscript by Walker et. al. explores the interplay between the global regulators HapR (the QS master high cell density (HDC) regulator) and CRP. Using ChIP-Seq, the authors find that at several sites, the HapR and CRP binding sites overlap. A detailed exploration of the murPQ promoter finds that CRP binding promotes HapR binding, which leads to repression of murPQ. The authors have a comprehensive set of experiments that paints a nice story providing a mechanistic explanation for converging global regulation. I did feel there are some weak points though, in particular the lack of integration of previously identified transcription start sites, the lack of replication (at least replication presented in the manuscript) for many figures, some oddities in the growth curve, and not reexamining their HapR/CRP cooperative binding model in vivo using ChIP-Seq.

Review of revised version:

This revised manuscript by Walker et. al. addresses some of the editorial points and conceptual discussion, but in general, most of my suggestions (as the previous reviewer #1) for additional experimentation or addition were not addressed as discussed below. Therefore, my overall review has not changed.

1. For example, in point 1, the suggested analysis was not performed because it is not trivial. My reason for making this suggestion is that the original manuscript was limited to Vibrio cholerae, and the impact of the manuscript would increase if the findings here were demonstrated to be more broadly applicable. I expect papers published in eLife to have such broad applicability. But no changes were made to the manuscript in this regard. The revised version is still limited to only *Vibrio cholerae*.

2. For point 2, the activity of FLAG-tag luxO could have been simply validated in a complementation assay. Yes, they demonstrated DNA binding, but that is not the only activity of LuxO.

3. For point 7, the transcriptional fusions were not explored at different times or different media, which is also something that was hinted at by other reviewers. In regard to exploring expression at different time points, this seems particularly relevant for QS regulated genes.

4. For point 13, the authors express that doing an additional CHIP-Seq is outside of the scope of this manuscript. Perhaps that is the case, but the point of the comment is to validate the in vitro binding results with an in vivo binding assay. A targeted CHIP-Seq approach specifically analyzing the promoters where cooperative binding was observed in vitro could have addressed this point.

---

## [Referee Report · Reviewer #2 (Public Review)]

This manuscript by Walker et al describes an elegant study that synergizes our knowledge of virulence gene regulation of *Vibrio cholerae*. The work brings a new element of regulation for CRP, notably that CRP and the high density regulator HapR co-occupy the same site on the DNA but modeling predicts they occupy different faces of the DNA. The DNA binding and structural modeling work is nicely conducted and data of co-occupation are convincing. The work seeks to integrate the findings into our current state of knowledge of HapR and CRP regulated genes at the transition from the environment and infection. The strength of the paper is the nice ChIP-seq analysis and the structural modeling and the integration of their work with other studies. The weakness is that it is not clear how representative these data are of multiple hapR/CRP binding sites or how the work integrates as a whole with the entire transcriptome that would include genes discovered by others. Overall this is a solid work that provides an understanding of integrated gene regulation in response to multiple environmental cues.

---

## [Author Response]

The following is the authors' response to the current reviews.

**Reviewer #1 (Public Review):**
This revised manuscript by Walker et. al. addresses some of the editorial points and conceptual discussion, but in general, most of my suggestions (as the previous reviewer #1) for additional experimentation or addition were not addressed as discussed below. Therefore, my overall review has not changed.

In our previous response, we included (i) extra experimental data illustrating the reproducibility of our results and (ii) added transcription start site data at the request of this reviewer. We included the information because we agreed with the reviewer that these were important points to address. For the points raised again below, we explained why the additional analysis was unlikely to add much in terms of insight or rigour. We have elaborated further below.

1. For example, in point 1, the suggested analysis was not performed because it is not trivial. My reason for making this suggestion is that the original manuscript was limited to Vibrio cholerae, and the impact of the manuscript would increase if the findings here were demonstrated to be more broadly applicable. I expect papers published in eLife to have such broad applicability. But no changes were made to the manuscript in this regard. The revised version is still limited to only *Vibrio cholerae*.

Our paper is focused on the unexpected co-operative interactions between HapR and CRP. Such co-binding of two transcription factors to the same DNA site is unexpected. Consequently, it is this mode of DNA binding that is likely to be of broad interest. With this in mind, we did provide experimental, and bioinformatic, analyses for other regulatory regions and other vibrio species (Figures S3 and S6). This, in our view, is where the “broad applicability” for papers published in eLife comes from.

The analysis the reviewer suggests is not related to the main message of our paper. Instead, the reviewer is asking how many HapR binding sites seen here by ChIP-seq are also seen in other vibrio species by ChIP-seq. This is only likely to be of interest to readers with an extremely specific interest in both vibrio species and HapR. The reviewer states above that we did not make the change “because it is not trivial”. This is an oversimplification of the rationale we presented in our response. The analysis is indeed not straightforward. However, much more importantly, the outcome is unlikely to be of interest to many readers, and has no bearing on the rigour of work. With this in mind, we do not think our position is unreasonable. We also stress that, should a reader with this very specific interest want to explore further, all of our data are freely available for them to do so.

1. For point 2, the activity of FLAG-tag luxO could have been simply validated in a complementation assay. Yes, they demonstrated DNA binding, but that is not the only activity of LuxO.

DNA binding by LuxO is the only activity of the protein with which we are concerned in our paper. Furthermore, LuxO is very much a side issue; we found binding to only the known targets and potentially, at very low levels, one additional target. No further LuxO experiments were done for this reason. Indeed, even if these data were removed completely, our conclusions would not change or be supported any less vigorously. We are happy to remove the LuxO data if the reviewer would prefer but this would seem like overkill.

1. For point 7, the transcriptional fusions were not explored at different times or different media, which is also something that was hinted at by other reviewers. In regard to exploring expression at different time points, this seems particularly relevant for QS regulated genes.

In their previous review, the reviewer did not request that such experiments were done. Similarly, no other reviewer requested these experiments. Instead, this reviewer (i) commented that lacZ fusions were not as sensitive as luciferase fusions (ii) asked if we had done any time point experiments. We agreed with the first point, whilst also noting that lacZ is not unusual to use as a reporter. For the second point, we responded that we had not done such experiments (which by the reviewer’s own logic would have been complicated using lacZ as a reporter). This seems like a perfectly reasonable way to respond.

We should stress that these comments all refer to Figure 2a, which was our initial screening of 23 promoter::lacZ fusions, supported by separate in vitro transcription assays. Only one of these fusions was followed up as the main story in the paper. Given that the other 22 fusions were not investigated further, and do not form part of the main story, there would seem little value in now going back to assay them at different time points.

1. For point 13, the authors express that doing an additional CHIP-Seq is outside of the scope of this manuscript. Perhaps that is the case, but the point of the comment is to validate the in vitro binding results with an in vivo binding assay. A targeted CHIP-Seq approach specifically analyzing the promoters where cooperative binding was observed in vitro could have addressed this point.

We did appreciate the original comment, and responded as such, but we do think additional ChIP-seq assays are outside the scope of this paper.

**Reviewer #2 (Public Review):**
This manuscript by Walker et al describes an elegant study that synergizes our knowledge of virulence gene regulation of *Vibrio cholerae*. The work brings a new element of regulation for CRP, notably that CRP and the high density regulator HapR co-occupy the same site on the DNA but modeling predicts they occupy different faces of the DNA. The DNA binding and structural modeling work is nicely conducted and data of co-occupation are convincing. The work seeks to integrate the findings into our current state of knowledge of HapR and CRP regulated genes at the transition from the environment and infection. The strength of the paper is the nice ChIP-seq analysis and the structural modeling and the integration of their work with other studies.

We thank the reviewer for the positive comments.

The weakness is that it is not clear how representative these data are of multiple hapR/CRP binding sites

This comment does not consider all data in our paper. We did test our model experimentally at multiple HapR and CRP binding sites. These data are shown in Figure S6 and confirm the co-operative interaction between HapR and CRP at 4 of a further 5 shared binding sites tested. We also used bioinformatics to show the same juxtaposition of CRP and HapR sites in other vibrio species (Figure S3). Hence, the model seems representative of most sites shared by HapR and CRP.

or how the work integrates as a whole with the entire transcriptome that would include genes discovered by others.

At the request of the reviewers, our revision integrated our ChIP-seq data with dRNA-seq data. No other suggestions to ingrate transcriptome data were made by the reviewers.

Overall this is a solid work that provides an understanding of integrated gene regulation in response to multiple environmental cues.

We thank the reviewer for the positive comment.

—————

The following is the authors' response to the original reviews.

**Reviewer #1 (Public Review):**
This manuscript by Walker et. al. explores the interplay between the global regulators HapR (the QS master high cell density (HDC) regulator) and CRP. Using ChIP-Seq, the authors find that at several sites, the HapR and CRP binding sites overlap. A detailed exploration of the murPQ promoter finds that CRP binding promotes HapR binding, which leads to repression of murPQ. The authors have a comprehensive set of experiments that paints a nice story providing a mechanistic explanation for converging global regulation.

We thank the reviewer for their positive evaluation.

I did feel there are some weak points though, in particular the lack of integration of previously identified transcription start sites

For completeness, we have now added the position and orientation or the nearest TSSs to each HapR or LuxO binding peak in Table 1 (based on Papenfort *et al*.).

the lack of replication (at least replication presented in the manuscript) for many figures,

We assume that the reviewer is referring to gel images rather than any other type of assay output (were error bars, derived from replicates, are shown). As is standard, we show representative gel images. All associated DNA binding and in vitro transcription experiments have been done multiple times. Indeed, comparison between figures reveals several instances of such replication (e.g. Figures 4b & 5d, Figures 4d & 5e). We have added details of repeats done to the methods section.

some oddities in the growth curve

We do not know why cells lacking *hapR* have a growth curve that appears biphasic. We can only assume that this is due to some regulatory effect of HapR, distinct from the *murQP* locus. Despite the unusual shape of the growth curve, the data are consistent with our conclusions.

and not reexamining their HapR/CRP cooperative binding model in vivo using ChIP-Seq.

We agree that these would be interesting experiments and, in the future, we may well do such work. Even without these data, our current model is well supported by the data presented (and the reviewer seems to agree with this above).

**Reviewer #2 (Public Review):**
This manuscript by Walker et al describes an elegant study that synergizes our knowledge of virulence gene regulation of *Vibrio cholerae*. The work brings a new element of regulation for CRP, notably that CRP and the high density regulator HapR co-occupy the same site on the DNA but modeling predicts they occupy different faces of the DNA. The DNA binding and structural modeling work is nicely conducted and data of co-occupation are convincing. The work could benefit from doing a better job in the manuscript preparation to integrate the findings into our current state of knowledge of HapR and CRP regulated genes and to elevate the impact of the work to address how bacteria are responding to the nutritional environment. Importantly, the focus of the work is heavily based on the impact of use of GlcNAc as a carbon source when bacteria bind to chitin in the environment, but absent the impact during infection when CRP and HapR have known roles. Further, the impact on biological events controlled by HapR integration with the utilization of carbon sources (including biofilm formation) is not explored.

We thank the reviewer for their overall positive evaluation.

The rigor and reproducibility of the work needs to be better conveyed.

Reviewer 1 made a similar comment (see above) and we have modified the manuscript accordingly.

Specific comments to address:

1. Abstract. A comment on the impact of this work should be included in the last sentence. Specifically, how the integration of CRP with QS for gene expression under specific environments impacts the lifestyle of Vc is needed. The discussion includes comments regarding the impact of CRP regulation as a sensor of carbon source and nutrition and these could be quickly summarized as part of the abstract.

We have added an extra sentence. However, we have used cautious language as we do not show impacts on lifestyle (beyond MurNAc utilisation) directly. These can only be inferred.

1. Line 74. This paper examines the overlap of HapR with CRP, but ignores entirely AphA. HapR is repressed by Qrrs (downstream of LuxO-P) while AphA is activated by Qrrs. With LuxO activating AphA, it has a significant sized "regulon" of genes turned on at low density. It seems reasonable that there is a possibility of overlap also between CRP and AphA. While doing an AphA CHIP-seq is likely outside the scope of this work, some bioinformatic or simply a visual analysis of the promoters known AphA regulated genes would be interest to comment on with speculation in the discussion and/or supplement.

In short, everything that the reviewer suggests here has already been done and was covered in our original submission (see text towards the end of the Discussion). Also, we would like to point the referee to our earlier publication (Haycocks *et al*. 2019. The quorum sensing transcription factor AphA directly regulates natural competence in *Vibrio cholerae*. *PLoS Genet.*
**15:**e1008362)*.*

1. Line 100. Accordingly with the above statement, the focus here on HapR indicates that the focus is on gene expression via LuxO and HapR, at high density. Thus the sentence should read "we sought to map the binding of LuxO and HapR of *V. cholerae* genome at high density".

Note that expression of LuxO and HapR is ectopic in these experiments (i.e. uncoupled from culture density).

1. Line 109. The identification of minor LuxO binding site in the intergenic region between VC1142 and VC1143 raises whether there may be a previously unrecognized sRNA here. As another panel in figure S1, can you provide a map of the intergenic region showing the start codons and putative -10 to -35 sites. Is there room here for an sRNA? Is there one known from the many sRNA predictions / identifications previously done? Some additional analysis would be helpful.

We have added an extra panel to Figure S1 showing the position of TSSs relative to the location of LuxO binding. We have altered the main text to accommodate this addition..

1. Line 117. This sentence states that the CHIP seq analysis in this study includes previously identified HapR regulated genes, but does not reveal that many known HapR regulated genes are absent from Table 1 and thus were missed in this study. Of 24 HapR regulated investigated by Tsou et al, only 1 is found in Table 1 of this study. A few are commented in the discussion and Figure S7. It might be useful to add a Venn Diagram to Figure 1 (and list table in supplement) for results of Tsou et al, Waters et al, Lin et al, and Nielson et al and any others. A major question is whether the trend found here for genes identified by CHIP-seq in this study hold up across the entire HapR regulon. There should also be comments in the discussion on perhaps how different methods (including growth state and carbon sources of media) may have impacted the complexity of the regulon identified by the different authors and different methods.

We have added a list of known sites to the supplementary material (new Table S1). We were unsure what was meant by the comment “A major question is whether the trend found here for genes identified by CHIP-seq in this study hold up across the entire HapR regulon”. We have added the extra comment to the discussion re growth conditions, also noting that most previous studies relied on in vitro, rather than in vivo, DNA binding assays.

1. The transcription data are generally well performed. In all figures, add comments to the figure legends that the experiments are representative gels from n=# (the number of replicate experiments for the gel based assays). Statements to the rigor of the work are currently missing.

See responses above. We have added a comment on numbers of repeats to the methods section.

1. Line 357-360. The demonstration of lack of growth on MurNAc is a nice for the impact of the work. However, more detailed comments are needed for M9 plus glucose for the uninformed reader to be reminded that growth in glucose is also impaired due to lack of cAMP in glucose replete conditions and thus minimal CRP is active. But why is this now dependent of hapR? A reminder also that in LB oligopeptides from tryptone are the main carbon source and thus CRP would be active.

We find this point a little confusing and, maybe, two issues (*murQP* regulation, and growth in general) are being conflated. In particular, we do not understand the comment “growth in glucose is also impaired due to lack of cAMP in glucose replete conditions and thus minimal CRP is active”.

Growth in glucose should indeed result in lower cAMP levels*, and hence less active CRP, but this does not impair growth. This is simply the cell’s strategy for using its preferred carbon source. If the reviewer were instead referring to some aspect of P_murQP_ regulation then yes, we would expect promoter activity to be lower because less active CRP would be available in the presence of glucose. The reviewer also comments “why is this now dependent of hapR?”. We assume that they are referring to some aspect of growth in minimal media with glucose. If so, the only hapR effect is the change in growth rate as cells enter mid-late log-phase (i.e. the growth curve looks somewhat biphasic). A similar effect is seen in all conditions. We do not know why this happens and can only conclude this is due to some unknown regulatory activity of HapR. Overall, the key point from these experiments is that loss if luxO, which results in constitutive hapR expression, lengthens lag phase only for growth with MurNAc as the sole carbon source.

*Although in *V. fischeri* (PMID: 26062003) cAMP levels increase in the presence of glucose.

1. A great final experiment to demonstrate the model would have been to show co-localization of the promoter by CRP and HapR from bacteria grown in LB media but not in LB+glucose or in M9+glycerol and M9+MurNAc but not M9+glucose. This would enhance the model by linking more directly to the carbon sources (currently only indirect via growth curves)

This is unlikely to be as straightforward as suggested. The sensitivity of CRP binding to growth conditions is not uniform across different binding sites. For instance, the CRP dependence of the *E. coli melAB* promoter is only evident in minimal media (PMID: 11742992) whilst the role of CRP at the *acs* promoter is evident in tryptone broth (PMID: 14651625). Similarly, as noted above, in *Vibrio fischeri* glucose causes and increase in cAMP levels. (PMID: 26062003).

1. Discussion. Comments and model focus heavily on GlcNAc-6P but HapR has a regulator role also during late infection (high density). How does CRP co-operativity impact during the in vivo conditions?

We really can’t answer this question with any certainty; we have not done any infection experiments in this work.

Does the Biphasic role of CRP play a role here (PMID: 20862321)?

Again, we cannot answer this question with any confidence as experimentation would be required. However, the suggestion is certainly plausible.

**Reviewer #3 (Public Review):**
Bacteria sense and respond to multiple signals and cues to regulate gene expression. To define the complex network of signaling that ultimately controls transcription of many genes in cells requires an understanding of how multiple signaling systems can converge to effect gene expression and ensuing bacterial behaviors. The global transcription factor CRP has been studied for decades as a regulator of genes in response to glucose availability. It's direct and indirect effects on gene expression have been documented in *E. coli* and other bacteria including pathogens including Vibrio cholerae. Likewise, the master regulator of quorum sensing (QS), HapR, is a well-studied transcription factor that directly controls many genes in Vibrio cholerae and other Vibrios in response to autoinducer molecules that accumulate at high cell density. By contrast, low cell density gene expression is governed by another regulator AphA. It has not yet been described how HapR and CRP may together work to directly control transcription and what genes are under such direct dual control.

We thank the reviewer for their assessment of our work.

Using both in vivo methods with gene fusions to lacZ and in vitro transcription assays, the authors proceed to identify the smaller subset of genes whose transcription is directly repressed (7) and activated (2) by HapR. Prior work from this group identified the direct CRP binding sites in the *V. cholerae* genome as well as promoters with overlapping binding sites for AphA and CRP, thus it appears a logical extension of these prior studies is to explore here promoters for potential integration of HapR and CRP. Inclusion of this rationale was not included in the introduction of CRP protein to the in vitro experiments.

We understand the reviewer’s comment. However, the rationale for adding CRP was not that we had previously seen interplay between AphA and CRP (although this is a relevant discussion point, which we did make). Rather, we had noticed that there was an almost perfect CRP site perfectly positioned to activate PmurQP. Hence, CRP was added.

Seven genes are found to be repressed by HapR in vivo, the promoter regions of only six are repressed in vitro with purified HapR protein alone. The authors propose and then present evidence that the seventh promoter, which controls murPQ, requires CRP to be repressed by HapR both using in vivo and vitro methods. This is a critical insight that drives the rest of the manuscripts focus. The DNase protection assay conducted supports the emerging model that both CRP and HapR bind at the same region of the murPQ promoter, but interpret is difficult due to the poor quality of the blot.There are areas of apparent protection at positions +1 to +15 that are not discussed, and the areas highlighted are difficult to observe with the blot provided.

We disagree on this point. The region between +1 and +15 is inherently resistant to attack by DNAseI and there are only ever very weak bands in this region (lane 1). Other than seeing small fluctuations in overall lane intensity (e.g. lanes 7-12 have a slightly lower signal throughout) the +1 to +15 banding pattern does not change. Conversely, there are dramatic changes in the banding pattern between around -30 and -60 (again, compare lane 1 to all other lanes). That CRP and HapR bind the same region is extremely clear. Also note that this is backed up by mutagenesis of the shared binding site (Figure 4c).

The model proposed at the end of the manuscript proposes physiological changes in cells that occur at transitions from the low to high cell density. Experiments in the paper that could strengthen this argument are incomplete. For example, in Fig. 4e it is unclear at what cell density the experiment is conducted.

Such details have been added to the figure legends and methods section.

The results with the wild type strain are intermediate relative to the other strains tested.

This is correct, and exactly what we would expect to see based on our model.

Cell density should affect the result here since HapR is produced at high density but not low density. This experiment would provide important additional insights supporting their model, by measuring activity at both cell densities and also in a luxO mutant locked at the high cell density. Conducting this experiment in conditions lacking and containing glucose would also reveal whether high glucose conditions mimicking the crp results.

We agree with this idea in principle but note that the output from our reporter gene, β- galactosidase, is stable within cells and tends to accumulate. This is likely to obscure the reduction in expression as cells transition from low to high cell density. Since we have demonstrated the regulatory effects of HapR and CRP both in vivo using gene knockouts, and in vitro with purified proteins, we think that our overall model is very well supported. Further experimental additions may provide an incremental advance but will not alter our overall story. Also note the unexpected increase in intracellular cAMP due to addition of glucose, in *Vibrio fischeri* (PMID: 26062003).

Throughout the paper it was challenging to account for the number of genes selected, the rationale for their selection, and how they were prioritized. For example, the authors acknowledged toward the end of the Results section that in their prior work, CRP and HapR binding sites were identified (line 321-22).

This is not quite what we say, and maybe the reviewer misunderstood, which is our fault. The prior work identified CRP sites whilst the current work identified HapR sites. We have made a slight alteration to the text to avoid confusion.

It is unclear whether the loci indicated in Table 1 all from this prior study. It would be useful to denote in this table the seven genes characterized in Figure 2 and to provide the locus tag for murPQ.

Again, we are unsure if we have confused the reviewer. The results in Table 1 are all HapR sites from the current work, not a prior study. However, some of these also correspond to CRP binding regions found in prior work.

The reviewer mentions “the seven genes characterised in Figure 2” but 23 targets were characterised in Figure 2a and 9 in Figure 2b. The “VC” numbers used in Figure 2 are the same as used in Table 1 so it is easy to cross reference between the two. We have added a footnote to Table 1, also referred to in the Figure 2 legend, to allow cross referencing between gene names and locus tags (including for *murQP* and *hapR*).

Of the 32 loci shown in Table 1, five were selected for further study using EMSA (line 322), but no rationale is given for studying these five and not others in the table.

This is not quite correct, we did not select 5 from the 32 targets listed in Table 1. We selected 5 targets from Table 1 that were also targets for CRP in our prior paper. This was the rationale.

Since prior work identified a consensus CRP binding motif, the authors identify the DNA sequence to which HapR binds overlaps with a sequence also predicted to bind CRP. Genome analysis identified a total of seven sites where the CRP and HapR binding sites were offset by one nucleotide as see with murPQ. Lines 327-8 describe EMSA results with several of these DNA sequences but provides no data to support this statement. Are these loci in Table 1?

This comment is a little difficult to follow, and we may have misunderstood, but we think that the reviewer is asking where the EMSA data referred to on lines 327-328 resides. We can see that the text could be confusing in this regard. We had referred to the relevant figure (Figure S6) on line 324 but did not again include this information further down in the description of the result. We have changed the text accordingly.

Using structural models, the authors predict that HapR repression requires protein-protein interactions with CRP. Electromobility shift assays (EMSA) with purified promoter DNA, CRP and HapR (Fig 5d) and in vitro transcription using purified RNAP with these factors (Figure 5e) support this hypothesis. However, the model proports that HapR "bound tightly" and that it also had a "lower affinity" when CRP protein was used that had mutations in a putative interaction interface. These claims can be bolstered if the authors calculate the dissociation constant (Kd) value of each protein to the DNA. This provides a quantitative assessment of the binding properties of the proteins.

The reviewer is correct that we do not explicitly provide a Kd. However, in both Figures 5d and 5e, we do provide very similar quantification. In 5d, our quantification is the % of the CRP-DNA complex bound by HapR (using either wild type or E55A CRP). Since the % of DNA bound is shown, and the protein concentrations are provided in the figure legend, information regarding Kd is essentially already present. In 5e, we show the % of maximal promoter activity. This is a reasonable way of quantifying the result. Furthermore, Kd is not a metric we can measure directly in this experiment that is not a DNA binding assay.

The concentrations of each protein are not indicated in panels of the in vitro analysis, but only the geometric shapes denoting increasing protein levels.

The protein concentrations are all provided in the figure legend. It is usual to indicate relative concentrations in the body of the figure using geometric shapes.

Panel 5e appears to indicate that an intermediate level of CRP was used in the presence of HapR, which presumably coincides with levels used in lane 4, but rationale is not provided.

There was no particular rationale for this, it was simply a reasonable way to do the experiment.

How well the levels of protein used in vitro compare to levels observed in vivo is not mentioned.

The protein concentrations that we use (in the nM to low μM range) are very typical for this type of work and consistent with hundreds of prior studies of protein-DNA interactions. The general rule of thumb is that 1000 molecules of a protein per bacterial cell equates to a concentration of around 1 μM. However, molecular crowding is likely to increase the effective concentration. Additionally, in vitro, where the DNA concentration is higher.

The authors are commended for seeking to connect the in vitro and vivo results obtained under lab conditions with conditions experienced by *V. cholerae* in niches it may occupy, such as aquatic systems. The authors briefly review the role of MurPQ in recycling of the cell wall of *V. cholerae* by degrading MurNAc into GlcNAc, although no references are provided (lines 146-50). Based on this physiology and results reported, the authors propose that murPQ gene expression by these two signal transduction pathways has relevance in the environment, where Vibrios, including *V. cholerae*, forms biofilms on exoskeleton composed of GlcNAc.

We have added a citation to the section mentioned.

The conclusions of that work are supported by the Results presented but additional details in the text regarding the characteristics of the proteins used (Kd, concentrations) would strengthen the conclusions drawn. This work provides a roadmap for the methods and analysis required to develop the regulatory networks that converge to control gene expression in microbes. The study has the potential to inform beyond the sub-filed of Vibrios, QS and CRP regulation.

As noted above, quantification essentially equivalent to Kd is already shown (% of bound substrate is indicated in figures and all protein concentrations are given in the figure legends).

**Reviewer #1 (Recommendations For The Authors):**
1. As similar experiments have been performed in other Vibrios, it would be interesting to do a more detailed analysis of the similarities and differences between the species. Perhaps a Venn diagram showing how many targets were found in all studies versus how many are species specific.

We appreciate this suggestion but would prefer not to make this change. A cross-species analysis would be very time consuming and is not trivial. The presence and absence of each target gene, for all combinations of organisms, would first need to be determined. Then, the presence and absence of binding signals for HapR, or its equivalent, would need to be determined taking this into account. For most readers, we feel that this analysis is unlikely to add much to the overall story. Given the amount of effort involved, this seems a “non-essential” change to make.

1. Line 101-Are the FLAG tagged versions of LuxO and HapR completely functional? Can they complement a luxO or hapR deletion mutant?

The activity of FLAG tagged HapR (LuxR in other Vibrio species) has been shown previously (e.g. PMIDs 33693882 and 23839217). Similarly, N-terminal HapR tags are routinely used for affinity purification of the protein without ill effect. We have not tested LuxO-3xFLAG for “full” activity, though this fusion is clearly active for DNA binding, the only activity that we have measured here, since all know targets are pulled down.

1. Line 106-As the authors state later that there are additional smaller peaks for HapR that could be other direct targets, I think a brief mention of the methodology used to determine the cutoff for the 5 and 32 peaks for LuxO and HapR, respectively, would be informative here.

We have added a little more text to the methods section. The added text states “Note that our cut- off was selected to identify only completely unambiguous binding peaks. Hence, weak or less reproducible binding signals, even if representing known targets, were excluded (see Discussion for further details)”.

1. Line 118-Need a reference here to the prior HapR binding site.

This has been added.

1. Figs. 1e-What do the numbers on the x-axis refer to? Why not just present these data as bases? The authors also refer to distance to the nearest start codon, but this is irrelevant for 4/5 of the luxO targets as they are sRNAs. They should really refer to the distance to the transcription start site. Likewise, for HapR, distance to the nearest start codon is not as informative as distance to the nearest transcription start site. A recent paper used transcriptomics to map all the transcription start sites of *V. cholerae*, and these results should be integrated into the author's study rather than just using the nearest start codon (PMID: 25646441).

The numbers are kilo base pairs, this has been added to the axis label. We have also changed “start codon” to “gene start” (since “gene start” is also suitable for genes that encode untranslated RNAs).

Re comparing binding peak positions to transcription start sites (TSSs) rather than gene starts, this analysis would be useful if TSSs could be detected for all genes. However, some genes are not expressed under the conditions tested by PMID: 25646441, so no TSS is found. Consequently, for HapR or LuxO bound at such locations, we would not be able to calculate a meaningful position relative to the TSS. We stress that the point of the analysis is to determine how peaks are positioned with respect to genes (i.e. that sites cluster near gene 5’ ends). Also note that nearest TSSs are now shown in the revised Table 1. In some cases, these are unlikely to be the TSS actually subject to regulation (e.g. because the regulated gene is switched off).

1. Fig. 1e-Is there directionality to the site? In other words, if a HapR binding site is located between two genes that are transcribed in opposite directions, is there a way to predict which gene is regulated? It looks like this might be the case with the list presented in Table 1, but how such determination is made and what the various symbol in Table 1 mean are not clear to me. This also has ramifications for Fig. 2a as the direction to construct the fusion is critical for the experiment.

The site is a palindrome so lacks directionality. The best prediction re regulation is likely to be positioning with respect to the nearest TSS (which is now included in Table 1). However, this would remain just a prediction and, where TSSs are in odd locations with respect to binding sites (taking into account the caveats above) predictions would be unreliable.

We are unsure which symbol the reviewer refers to in Table 1, a full explanation of any symbols used is provided in the table footnotes.

With respect to Figure 2a, if sites were between divergent genes, and met our other criteria, we tested for regulation in both directions. For example, see the divergent genes VCA0662 (classified inactive) and VCA0663 (classified repressed).

1. Fig. 2a-It is a little disappointing that the authors use LacZ fusions to measure transcription as this is not the most sensitive reporter gene. Luciferase gene fusions would have been much more sensitive. Also, did the authors examine multiple time points. The methods only describe "mid-log phase" but some of the inactive promoters could be expressed at other time points. Also, it would be simple to repeat this experiment in different media, such as minimal with glucose or another non- CRP carbon source, to expand which promoters are expressed.

The reviewer is correct regarding the sensitivity of β-galactosidase, which is very stable and so accumulates as cells grow. Even so, this reporter has been used very successfully, across thousands of studies, for decades. We did not examine multiple timepoints. We appreciate that the 23 promoter::lacZ fusions could be re-examined using varying growth conditions but this is unlikely to impact the overall conclusions, though it could generate some new leads for future work.

1. Fig. 2a legend-typos

This has been corrected.

1. Line 138-I think you mean Fig. 2a here.

This has been corrected.

1. Fig. 2b and many additional figures quantify band intensity but do not show any replication or error. Therefore, it is impossible to gauge reproducibility of these experiments.

We have added a reproducibility statement (all experiments were done multiple times with similar results) as is standard throughout the literature. Also note that there is a lot of internal replication between figures. Figure 4d and Figure 5e lanes 1-9 show essentially the same experiment (albeit with slightly different protein concentrations) and very similar results. To the same effect, Figure 5e lanes 10-18 and lanes 19-27 show the same experiment for two different mutations of the same CRP residue. Again, the results are very similar. Also see the response to your comment 15 below.

1. Fig. 4a-lanes 2-4-the footprint does not change with additional CRP. In other words, it looks the same at the lowest concentration of CRP versus the highest concentration of CRP. The footprints for HapR look similar. This is somewhat troubling as in these types of experiments one would like to observe a dose dependent change in the footprint correlating with more DNA occupancy.

For CRP we agree but are not concerned at all by this. The site is simply full occupied at the lowest protein concentration tested. Given that the footprint exactly coincides with a near consensus CRP site (which, when mutated, abolishes CRP binding in EMSAs, and regulation by CRP in vivo) all our results are perfectly consistent. Note that (i) our only aim in this experiment was to determine the positions of CRP and HapR binding (ii) our conclusions are independently backed up using gel shifts and by making promoter mutations. With respect to HapR, there are changes at the periphery of the main footprint.

1. Fig. 4e-Why does the transcriptional activation of murQP decrease with increasing concentrations of CRP? This is also seen in Fig. 5e.

In our experience, this often does happen when doing *in vitro* transcription assays (with CRP and many other activators). The anecdotal explanation is that, at higher concentrations, the regulator can start to bind the DNA non-specifically and so interfere with transcription.

13. The authors demonstrate in vitro that HapR requires binding of CRP to bind the murQP promoter. It would strengthen their model if they demonstrated this in vivo. To do this, the authors only need to repeat their ChIP-Seq experiment in a delta CRP mutant and the HapR signal at murQP would be lost. In fact, such an experiment would experimentally confirm which of the in vivo HapR binding sites are CRP dependent.

We agree, appreciate the comment, and do plan to do such experiments in the future as a wider assessment of interactions between transcription factors. However, doing this does have substantial time and resource implications that we cannot devote to the project at present. We feel that our overall conclusions, regarding co-operative interactions between HapR and CRP at PmurQP, are well supported by the data already provided. This also seems the overall opinion of the reviewers.

1. Fig. 5b-I am confused by the Venn diagram. The text states that "In all cases, the CRP and HapR targets were offset by 1 bp", but the diagram only shows 7 overlapping sites. The authors need to better describe these data.

We mean that, in all cases where sites overlap, sites are offset by 1 bp (i.e. we didn’t find any sites overlapping but offset by 2, 3 4 bp etc).

15. Line 287-288 and Fig. 5d-The authors state that HapR binds with less affinity to the CRP E55A mutant protein bound to DNA. There does seem to be a difference in the amount of shifted bands at the equivalent concentrations of HapR, but the difference is subtle. In order to make such a conclusion, the authors should show replication of the data and calculate the variability in the results. The authors should also use these data to determine the actual binding affinities of HapR to WT and the E55A mutant CRP, along with error or confidence intervals.

All of these experiments have been run multiple times and we are absolutely confident of the result. With respect to Figure 5d, this was done many times. We note that not all experiments were exact repeats. E.g. some of the first attempts had fewer HapR concentrations. Even so, the defect in HapR binding to the CRP E55A complex was always evident. The two gels to the left show the final two iterations of this experiment (these are exact repeats). The top image is that shown in Figure 5d. The lower image is an equivalent experiment run a day or so previously. Both clearly show a defect in HapR binding to the CRP E55A complex. We appreciate that our conclusion re these experiments is somewhat qualitative (i.e. that HapR binds the CRP E55A complex less readily) but this is not out of kilter with the vast majority of similar literature and our results are clearly reproducible.

1. Fig. 6a-It is odd that the locked low cell density mutants have such a growth defect in MurNAc, minimal glucose, and LB. To my knowledge, such a growth defect is not common with these strains. Perhaps this has to do with the specific growth conditions used here, but I can't find that information in the manuscript (it should be there). Furthermore, the growth rate of the luxO and hapR mutants appears to be similar up to the branch point (i.e. slope of the curve), but the lag phage of the luxO mutant is much longer. The authors need to address these issues in relationship to previous published literature and specify their growth conditions because the results are not consistent with their simple model described in Fig 6b.

This comment is a little difficult to pick apart as it covers several different issues. We’ll try and

answer these individually.

a) The unusual “biphasic growth curve with hapR and hapRluxO cells: We do not know why cells lacking hapR have a growth curve that appears biphasic. We can only assume that this is due to some regulatory effect of HapR, distinct from the *murQP* locus. Despite the unusual shape of the growth curve, the data are consistent with our conclusions.

b) The extended lag phase of the luxO mutant in minimal media + MurNAc: We appreciate this comment and had considered possible explanations prior to submission. In the end, we left out this speculation but are happy to include it as part of our response. The extended lag phase might be expected if CRP/HapR regulation is largely critical for controlling the basal transcription of murQP. The locus is likely also regulated by the upstream repressor MurR (VC0204) as in *E. coli*. So, if deprepression of MurR overwhelms the effect of HapR on murQP, we think you would expect that once the cells start growing on MurNAc, the growth rates are unchanged. But the extended lag is due to the fact that it took longer for those cells to achieve the critical threshold of intracellular MurNAc-6-P necessary to drive murR derepression. Obviously, we can not provide a definitive answer.

c) We have added further details regarding growth conditions to the methods section and the Figure 6a legend.

1. Fig. S6-The data to this point with murPQ suggested a model in which CRP binding then enabled HapR binding. But these EMSA suggest that both situations occur as in some cases, such as VCA0691, HapR binding promotes CRP binding. How does such a result fit with the structural model presented in Fig. 5?

This is to be expected and is fully consistent with the model. Cooperativity is a two-way street, and each protein will stabilise binding of the other. Clearly, it will not always be the case that the shared DNA site will have a higher affinity for CRP than HapR (as at PmurQP). Depending on the shared site sequence, expected that sometimes HapR will bind “first” and then stabilise binding of CRP.

18. Line 354-356-The HCD state of *V. cholerae* occurs in mid-exponential phase and several cell divisions occur before stationary phase and the cessation of growth, at least in normal laboratory conditions. Therefore, there is not support for the argument that QS is a mechanism to redirect cell wall components at HCD because cell wall synthesis is no longer needed.

We did not intent to suggest cell wall synthesis is not needed at all, rather that there is a reduced need. We made a slight change to the discussion to reflect this.

19. Line 357-360-Again, as stated in point 16, the statement that cells locked in the HCD are "defective for growth" is an oversimplification. The luxO mutants have a longer lag phage, but they actually outgrow the hapR mutants at higher cell densities and reach the maximum yield much faster.

In fairness, we do go on to specify that the defect is an extended lag phase. Also see our response above.

**Reviewer #2 (Recommendations For The Authors):**
Comments to improve the text1. Line 103-106, line 130, line 136, etc. Details of the methods and the text directing to presentations of figures should be in the methods and/or figure legends with (Figure x) in citation after the statement. The sentences in lines indicated can be deleted from the results. Although several lines are noted specifically here, this comment should be applied throughout the entire results section.

We appreciate this comment but would prefer not to make this change (it seems mainly an issue of personal stylistic choice). It is sometimes helpful for the reader to include such information as it avoids them having to cross reference between different parts of the manuscript.

1. Line 115. Recommend a paragraph between content on LuxO and HapR (before "Of the 32 peaks for HapR binding")

We agree and have made this change.

1. Line 138 and Figure 1a. I am not convinced this gel shows that VC1375 is activated by HapR. Is the arrow pointing to the wrong band? There does seem to be an induced band lower down.

We understand this comment as it is a little difficult to see the induced band. This is because this is a compressed area of the gel and the transcript is near to an additional band.

1. Line 147. Add the VC0206-VC0207 next to murQP (and the gene name murQP into Table 1).

We have added the gene name to the figure foot note. The text has been changed as requested.

1. Methods. It is essential for this paper to have detailed methods on the bacterial growth conditions. Referring to prior paper, bacteria were grown in LB (add composition...is this high salt LB often used for vibrios or low salt LB often used for *E. coli*). Growth is to "mid log". Please provide the OD at collection. Is mid log really considered "high density". Provide a reference regarding HapR activity at mid log to support the method. Could the earlier collection of bacteria account for missing known HapR regulated genes? In preparing the requested ç, include growth conditions for other experiments in the legends.

Note that we have included a new supplementary table, rather than a Venn diagram. We have also added further details of growth conditions as mentioned above. Also not that, for the ChIP-seq, HapR and LuxO were expressed ectopically and so uncoupled from the switch between low and high cell density.

1. Content of Table 1, HapR Chip-seq peaks, needs to be closely double checked to the collected data as there seems to be some errors. Specifically, VC0880 and VC0882 listed under Chromosome I are most likely VCA0880 (MakD) and VCA0882 (MakB), both known HapR induced genes on Chromosome II with VCA0880 previously validated by EMSA. This notable error suggests the table may have other errors and thus requires a very detailed check to assure its accuracy.

We appreciate the attention to detail! We have double checked, thankfully this is not an error, the table is correct (even so, we have also checked all other entries in the table). As an aside, VCA0880 is one of the locations for which we see a weak HapR binding signal below our cut-off (included in the new Table S1). In cross checking between Table 1 and all other data in the paper we noticed that we had erroneously included assay data for VC0620 in Figure 2A. This was not one of our ChIP-seq targets but had been assayed at the same time several years ago. This datapoint, which wasn’t related to any other part of the manuscript, has been removed.

If VCA0880 and VCA0882 are correctly placed on Chr. I, then add comment to text that the Mak toxin genomic island found on Chromosome II in N16961 is on Chr. I in E7946. (See recent references PMID: 30271941, 35435721, 36194176, 34799450).

See above, this is not an error.

1. Alternatively for both comments 8 & 9, are these problems of present/missing genes or misannotations the result of the annotation of E7946 gene names not aligning with gene names of N16961? (if so, in Table 1, please give the gene name as in E7946 but include a separate column with the N16961 name for cross study comparison)

See above and below, this is not an issue.

1. Line 126-127. Also regarding Table 1, please add a column with function gene annotation. For example, VC0916 needs to be identified as vpsU. If function is unknown, type unknown in the column. This will help validate the approach of selecting "HapR target promoters where adjacent coding sequence could be used to predict protein function."

We added an extra column to Table 1 in response to a separate reviewer request (TSS locations). This leaves no space for any additional columns. Instead, to accommodate the reviewer’s request, we have added alternative gene names to the footnote.

Not following up on VCA0880 (promoter for the mak operon) is a sad missed opportunity here as it is one of the most strongly upregulated genes by HapR (PMC2677876)

As noted above, this was not an error and VCA0880 was not one of our 32 HapR targets. As such, we would not have followed this up.

1. Figure Legends. Add a unit to the bar graphs in Figure 1e (should be kb??) This has been corrected.2. The yellow color text labels in figures 3c, 4a, 4c are difficult to read. Can you use an alternative darker color for CRP.

We have made this slightly darker (although to our eye it is easily reliable). We haven’t changed the colour too much, for consistency with colour coding elsewhere.

1. Figure S3. Binding is misspelled. Add units to the x-axis

This has been corrected.

1. Figure S7. The text in this figure is too small to read. Figure could be enlarged to full page or text enlarged. Are these 4 the only other known regulated promoters? Could all the known alternative promoters linked to HapR be similarly probed?

We have increased the font size and included a new Table S1 for all previously proposed HapR sites.

1. Figure S8. Original images..are any of these the replicate gels (see public comment 6)

We have added a statement regarding reproducibility, and also note the internal reproducibility between different figures in our reviewer response. The gels in Figure S8 are full uncropped versions of those shown in the main figures.

**Reviewer #3 (Recommendations For The Authors):**
None

Whilst there were no specific recommendations from this reviewer, we have still responded to the public review and made changes if required.